# Highly stressful global event affecting health sciences students: A longitudinal qualitative study

Yolanda E. Salazar-Granizo [1,2,3]*, Rafael A. Caparros-Gonzalez[2,4], Daniel Puente-Fernandez[2,4], César Hueso-Montoro[5,6]

1 Doctorate Program in Clinical Medicine and Public Health, University of Granada, Granada, Spain, 2 Instituto de Investigación Biosanitaria ibs, GRANADA, Granada, Spain, 3 Carrera de Enfermería, Facultad de Ciencias de la Salud, Universidad Nacional de Chimborazo, Riobamba, Ecuador, 4 Department of Nursing, Faculty of Health Sciences, University of Granada, Granada, Spain, 5 Department of Nursing, Faculty of Health Sciences, University of Jaen, Jaen, Spain, 6 Center for Mind, Brain, and Behavior Research (CIMCYC), Granada, Spain

☯ These authors contributed equally to this work.
* ysalazarg@unach.edu.ec, ysalazar@correo.ugr.es

## Abstract

### Background

Stressful events of great magnitude have produced significant changes in society and in health education. University students have faced considerable challenges both during and after such events, which have affected their lifestyles, mental health, the development of academic activities, and changes in education systems due to the adoption of new teaching models and the use of online technology.

### Objectives

To explore the perspectives and experiences of university health sciences students regarding their lifestyles and academic stress both during and after the COVID-19 pandemic, a highly stressful event. This study also aimed to establish relationships among the criteria involved.

### Design and participants

Qualitative analytical-interpretative research using the hermeneutic method was conducted, and the coded numerically matched responses were complementarily analyzed with inferential statistics. The sample consisted of 1,735 students enrolled in the Nursing, Physiotherapy, Clinical Laboratory, Medicine, Dentistry, and Clinical Psychology programs of the Faculty of Health Sciences at the National University of Chimborazo in Ecuador. Data were collected considering two time points: during mandatory social isolation (T1 = virtual modality) and upon the return to face-to-face activities (T2 = face-to-face modality).

**Data availability statement:** The anonymized database has been included in: https://figshare.com/s/dc53fc6cf768800383d5.

**Funding:** National University of Chimborazo (doctoral studies grant 0250-CU-UNACH-SE-ORD-17-08-2022). The funders had no role in study design, data collection and analysis, decision to publish, or preparation of the manuscript.

**Competing interests:** NO authors have competing interests.

## Results

After analyzing the texts produced by the students, four main themes were identified: (1) lifestyle modifications; (2) the alteration of academic activities; (3) a preference for the vocational training modality; and (4) academic stress. Significant differences were identified between male and female students. As for academic programs with numerically matched codes 33 codes were identified at T1 and 35 at T2.

## Conclusions

COVID-19, a stressful event of great magnitude impacted the lifestyles of health sciences students and caused them academic stress. Academic program and gender were statistically significant in some of the changes. During isolation and in the return to face-to-face learning, the students modified their lifestyles and experienced academic overload.

## Introduction

Stressful events that affect academics, families, society, work, or even large-scale events that affect an entire population can influence health in addition to well-being [1], generating an unprecedented impact on society; health education is no exception [2]. Students in particular face significant challenges during and after stressful events, which affect their lifestyles and mental health [3,4]. Further, changes take place in education systems due to the adoption of new teaching models and the use of online technology [5], with the aim of mitigating the consequences of stressful events.

A healthy lifestyle is a simple and safe way to delay the emergence of disease [6]. During periods of isolation that are enforced in response to an adverse event, lifestyles are negatively influenced [7], leading to physical inactivity, an unhealthy diet, weight gain, neck and shoulder pain, difficulty sleeping, anxiety, and worries, with a higher prevalence of these phenomena occurring in women [8].

Health education underwent a major transition due to the COVID-19 pandemic, especially changes in education models [9]. Students exhibited uncertainty due to ineffective learning, low motivation, impaired socialization, and reduced communication [10]; these problems, which arose due to digital instruction, are limitations of academic distance learning [11]. In addition, there is concern about clinical training, as encounters with patients allow for the development of clinical acumen, technical ability, and experience [12]. Limitations during online education especially in hands-on activities, internships, labs, and simulations as well as potential negative effects on academic performance increase stress levels during social isolation [11,13].

After a major stressful event, people must choose to thrive rather than simply survive [14]. The demonstrated ability to adapt to multiple challenges is an indicator of a shift in behavior and requires not returning to one's previous routines [15]. Moreover, universities must respond from an educational perspective by using alternative resources to compensate for incomplete or lost training [16,17]. The future

must be planned for by reflecting on strengths, needs, and deficits, mitigating the negative effects perceived resulting from changes in learning modality [5].

Higher education institutions must adapt to the time following a stressful event and innovate their activities, offering health sciences students opportunities to achieve learning outcomes [18]. The educational response that must be applied to mitigate unfavorable effects should include deepening the perspectives and experiences of university health sciences students regarding their lifestyles and academic stress both during and after stressful events.

Thus, it is essential to identify the views and experiences of health sciences students regarding their lifestyles and academic stress during and after a stressful event of great magnitude such as the COVID-19 pandemic.

## Method

### Design

In the present qualitative study, which has an analytical-interpretative nature, used the hermeneutic method [19], which allowed us to explore the students' perspectives and experiences [20] both during and after the COVID-19 pandemic. Subsequently, the relationships between the criteria involved were established. The coded responses, numerically matched, were complementarily analyzed with inferential statistics, which allowed us to investigate the relationships between the variables of academic program and sex with the codes. This effort supported the qualitative analysis.

Information collection considering two time points: time 1 (T1): the period of the online learning modality from April 18, 2022, to August 4, 2022 (academic period 2022, 1S); and time 2 (T2): the return of students to face-to-face activities between November 7, 2022, and March 10, 2023 (academic period 2022, 2S).

### Study population (Recruitment)

The sample consisted of students (N = 2,880) enrolled in the academic programs of Nursing, Physiotherapy, Clinical Laboratory, Medicine, Dentistry, and Clinical Psychology of the Faculty of Health Sciences at the National University of Chimborazo in Ecuador). The sample was selected through non-probabilistic sampling for convenience, the accessibility of the student population and the importance of the maximum number of relevant individuals participating at both time points was considered; 1,735 responses were obtained of students who were legally enrolled during the T1 (April 18, 2022, to August 4, 2022) and T2 (November 7, 2022, and March 10, 2023).

To collect data for T1, questionnaires were administered through the university's Academic Control System (SICOA) following authorization from academic authorities in July 2023. These questionnaires were accessible from August 1 to September 30, 2023 (recruitment period T1). For T2 data collection, questionnaires were again administered via SICOA after obtaining authorization from academic authorities in October 2023. The T2 questionnaires were available from November 1, 2023, to February 29, 2024 (recruitment period T2).

The inclusion criteria were (1) being enrolled in an academic health sciences program at T1 and T2; (2) being over 18 years of age; and (3) answering all questions for both moments (T1 and T2). The exclusion criteria were (1) not having an active enrollment in T1 y T2, (2) being under 18 years of age, (3) not having answered the complete questionnaires for T1 and T2, thus ensuring data consistency.

The students received Information regarding the study's objectives, methodological approach, potential risks, and benefits the investigation, in the document "Information to the Participant" envoy through e-mail, in addition, synchronous presentations were made by virtual means of the authors to provide additional information about the research, clarify doubts, solve concerns and motivate the participation of the students. Considering the health measures imposed (isolation, social distancing, among others) during the Covid-19 pandemic, students who voluntarily agreed to participate in the study issued their informed digital consent prior to accessing and filling out the questionnaires in the Academic Control Computer System of the University (online).

## Ethical considerations

This article is part of the results of the research "Lifestyle and academic stress in Health Sciences students in an Ecuadorian educational environment" was approved by the Human Research Ethics Committee of the Catholic University of Cuenca in Ecuador (Resolution #CEISH – UCACUE – 052). The committee is recognized by Ecuador's Ministry of Public Health. A structured digital form, implemented in the SICOA system, facilitated the acquisition of digital informed consent from each participant, adapting the traditional written process to an online format due to the social distancing measures in effect at the time. Participants indicated their consent or non-consent through mandatory selection of a checkbox, which automatically recorded the timestamp of selection and stored it in a secure database.

If students provided consent, they proceeded to complete the research forms. The study excluded participants under 18 years of age.

In this study, the confidentiality of data provided by participants was prioritized. Consequently, an anonymization procedure was implemented to safeguard confidentiality, effectively minimizing the risk of individual participant identification. Information regarding the study's objectives, methodological approach, potential risks, and anticipated benefits was conveyed to each participant prior to obtaining their informed consent, thereby ensuring the integrity of the research process.

The entire investigation was conducted in strict accordance with fundamental bioethical principles: non-maleficence, beneficence, autonomy, and justice. By rigorously adhering to these principles throughout all phases of the study, from conception to completion, the research team ensured full compliance with the ethical guidelines and regulatory requirements governing human subject research in the scientific community.

## Instruments

To collect data, a questionnaire was created with eight open-ended questions that were previously validated by specialists with recognized teaching and research profiles in the field of health and aimed to understand health sciences students' perspectives on their lifestyles and level of academic stress in a stressful situation. On the questionnaire administered for the period the COVID-19 pandemic, four questions inquired about significant aspects of the defined themes: (1) students' perceptions regarding how the development of virtual academic activities affected their lifestyle due to social isolation; (2) how these modifications influenced their academic performance and their preference for a certain modality (face-to-face, virtual, or hybrid); and (3) the arguments of their choice. Finally, the students were quested about the (4) impact of social isolation on the generation of academic stress.

To the return to academic activities in the face-to-face mode after the pandemic, once social isolation ended, a questionnaire with four open-ended questions was administered. The impact of the return to face-to-face academic activities performance and the preference for modality was investigated. Finally, it was explored how the return to the face-to-face teaching affected the generation of academic stress in students.

In addition, demographic information was collected (i.e., age, sex, nationality, self-identification, marital status, economic dependence, the student's academic program, level of study, and academic average on a scale of zero to ten, and this information was classified in accordance with current national higher education regulations.

## Data collection procedure

Data collection involved students from six academic programs in the Faculty of Health who were enrolled during two distinct periods: Time 1 (T1): Students enrolled April-August 2022, data collection period: August-September 2023 Time 2 (T2): Students enrolled November 2022-March 2023, data collection period: November 2023-February 2024. All data were collected through a structured digital questionnaire with open-ended questions administered via the university's Academic Control System (SICOA), obtaining online narrative responses.

Given the pandemic-related restrictions and mandatory social distancing both during and after the COVID-19 pandemic, a virtual approach was implemented to invite students to participate through email with the submission of a

participant information document. Also, through online platforms, synchronous virtual activities were organized to communicate the study's objectives and project scope, emphasize volunteer participation, and encourage students to answer questions.

With the necessary approvals, the data was collected online considering two time points: (1) during mandatory social isolation in the virtual learning modality; and (2) the return to face-to-face activities.

To protect the confidentiality of the participants and to identify those who took filled out the polls at both times, an anonymization process was used, by generating a numerical code for each student, thus reducing the probability that they would be identified and allowing us to link answers to both time points.

The first author is a professor in one of the university's academic health programs, which provides a valuable institutional perspective on the students' experiences. The research team was composed of members with extensive experience in qualitative methods that constitute a solid basis for the present study.

The research team, aware of the influence of their own experiences both during and after the COVID-19 pandemic on students' perceptions, incorporated measures for participant validation and data triangulation to ensure that students' opinions would be accurately represented in the findings.

## Data analysis

Once the information was collected, the participants' answers were reviewed and selected those who completed all questions for both time points, obtaining a sample of 1,735 students.

To analyze the experiences expressed in the textual data, subjective interpretation of the content of textual data developed by Miles and Huberman (1994) [21] was used. This method includes classification, codification, and identification of themes to understand and generalize the participants' perceptions.

We identified and grouped the participants' ideas into themes, categories, and codes during T1 and T2, considering concurrence with those mentioned in their opinions, following the guidelines of Briand et al. [22] to guarantee the quality of the process. The texts were analyzed using the qualitative software ATLAS.ti, which is considered robust for handling a large volume of unstructured data. In addition to its capacity for complex analysis of qualitative content, this tool allows for the visualization of conceptual networks and cooccurrence analysis [23]. Moreover, the inductive analysis, which was based on the students' shared experiences, permitted us to understand different aspects of their views.

To ensure the reliability and validity of the findings, a researcher with a master's degree and PhD candidate in health thoroughly reviewed the participants' criteria. Guided by reflexivity, recognized previous assumptions, coded the data, and suggested categories. The research team, composed of members with a PhD background and vast experience in qualitative research, discussed the coherence of the coding, the acceptance of categories, and the consolidation of themes until reaching a consensus, as mentioned by Motlagh et al. [24]. This approach allowed us to interpret the results to be validated and enriched, thus guaranteeing the reliability of the study. In addition, an audit trail has been maintained so that all research documentation will be accessible for future consultation [25].

Descriptive data are presented as frequencies and measures of central tendency. The codes presented as numerical data allowed for an inferential analysis of two variables (academic program and sex) via chi-square ($\chi^2$) tests as a complement to the qualitative research. The quantitative processing of the data was carried out using the SPSS Statistical Package for Social Sciences, version 24.0 (IBM Corporation: Armonk, NY, US). This combination allowed for a more coherent, enriched interpretation of the results, maximizing the explanatory potential of the research.

To analyze the volume of responses (n = 1,735 participants; 8 questions; 2 time points), we implemented a systematic analytical approach that maintained qualitative rigor while effectively managing the extensive dataset, as follows:

ATLAS.ti Analysis: Initial Coding Phase:

• All textual responses were imported into ATLAS.ti

• Data were organized by time points (T1 and T2) and by question

- Coding was conducted following qualitative content analysis principles

- Codes were iteratively refined through research team discussions

- Code groups were established based on emerging themes

- Network views were created to visualize relationships between codes

   Code Analysis and Quantification Process

- Codes were analyzed independently for each question and assigned a number according to their presentation by question and time

SPSS Transition:

- Code numbers were organized by question and time and appended to the demographic database, matching responses by participant according to the assigned anonymized code

- Chi-square tests were conducted to analyze relationships between:

  ◦ Code presence in responses by question and theme

  ◦ Academic programs

  ◦ Participants' sex

This approach maintained the integrity of the qualitative analysis while enabling pattern exploration through statistical analysis, complementary to the primary qualitative findings, by identifying possible relationships between coded responses and demographic variables, as evidenced in the tables presented in the results section."

## Results

### Descriptive findings

The participants included 1,259 (72.6%) women and 476 (27.4%) men. Their mean age was 21.15 years (SD = 2.32) at T1 and 22.78 years (SD = 2.37) at T2. Most of them were Ecuadorian (1,722, 99.3%), self-identified as mestizo (1,587, 91.5%), and were single (1,692, 97.5% at T1 and 1,682, 96.9% at T2). They were economically dependent on their parents more frequently at T1 (1,583, 91.2%) than at T2 (1,356, 78.2%). The mean academic average was slightly greater at T1 (8.28, SD = 0.71) than at T2 (8.12, SD = 0.72) on a scale of 0–10, with an increase in T2 among students who failed from 56 (3.2%) at T1 to 109 (6.30%) at T2 (Table 1).

### Analysis of lifestyles, academic stress, and preference for modality

The perspectives of health sciences students during the COVID-19 pandemic were obtained by analyzing the collected data. Four main themes, two categories, and 68 codes were identified during the T1 (April–August 2022) and T2 (November–March 2023) time periods. The themes were determined based on attendance and frequency with which the participants mentioned them in their opinions. The categories correspond to the time in which they were presented (Category 1: mandatory social isolation, virtual learning modality; Category 2: return to face-to-face activities).

   The codes corresponding to each theme and category were selected according to their repetition in the students' responses with mentions on 10 or more occasions.

   The percentages presented in Tables 2–5 reflect the share of students who mentioned each code by sex in relation to the total number of participants in their respective programs; the total percentages reflect the proportion in relation to the overall sample (Fig 1).

**Table 1. Data demographics at T1 and T2, Health Students (n = 1735).**

| Variable | First Moment T1 | | | Second Moment T2 | | |
|---|---|---|---|---|---|---|
| Age* | Frequency | % | M (SD) | Frequency | % | M (SD) |
| = 18 | 151 | 8.7 | | | | |
| 19–25 | 1525 | 87.9 | 21.15 (2.32) | 1579 | 91 | 22.78 (2.37) |
| 26–32 | 53 | 3.1 | | 147 | 8.5 | |
| 33–39 | 4 | .2 | | 7 | .4 | |
| 40+ | 2 | .1 | | 2 | .1 | |
| **Sex** | | | | | | |
| Female | 1259 | 72.6 | | 1259 | 72.6 | |
| Male | 476 | 27.4 | | 476 | 27.4 | |
| **Nationality** | | | | | | |
| Ecuadorian | 1722 | 99.3 | | 1722 | 99.3 | |
| Colombian | 2 | .1 | | 2 | .1 | |
| Cuban | 1 | .1 | | 1 | .1 | |
| American (USA) | 1 | .1 | | 1 | .1 | |
| Spanish | 3 | .2 | | 3 | .2 | |
| Venezuelan | 6 | .3 | | 6 | .3 | |
| **Identification** | | | | | | |
| Indigenous | 120 | 6.9 | | 120 | 6.9 | |
| White | 8 | .5 | | 8 | .5 | |
| Mestizo | 1587 | 91.5 | | 1587 | 91.5 | |
| Afro-Ecuadorian | 8 | .5 | | 8 | .5 | |
| Montubio | 6 | .3 | | 6 | .3 | |
| Other | 6 | .3 | | 6 | .3 | |
| **Marital status** | | | | | | |
| Single | 1692 | 97.5 | | 1682 | 96.9 | |
| Married | 24 | 1.4 | | 37 | 2.1 | |
| Divorced | 3 | .2 | | 6 | .3 | |
| Cohabitating | 16 | .9 | | 10 | .6 | |
| **Economic dependence** | | | | | | |
| Not applicable | 57 | 3.3 | | 328 | 18.9 | |
| Parents | 1583 | 91.2 | | 1356 | 78.2 | |
| Family | 71 | 4.1 | | 39 | 2.2 | |
| Couple | 15 | .9 | | 11 | .6 | |
| Other | 9 | .5 | | 1 | .1 | |
| **Academic program** | | | | | | |
| Nursing | 223 | 12.9 | | 223 | 12.9 | |
| Medicine | 430 | 24.8 | | 430 | 24.8 | |
| Physical therapy | 231 | 13.3 | | 231 | 13.3 | |
| Clinical laboratory | 204 | 11.8 | | 204 | 11.8 | |
| Dentistry | 393 | 22.7 | | 393 | 22.7 | |
| Clinical psychology | 254 | 14.6 | | 254 | 14.6 | |
| **Level** | | | | | | |
| First | 408 | 23.5 | | 268 | 15.4 | |
| Second | 180 | 10.4 | | 177 | 10.2 | |
| Third | 221 | 12.7 | | 179 | 10.3 | |
| Fourth | 285 | 16.4 | | 233 | 13.4 | |

*(Continued)*

**Table 1.** (Continued)

| Variable | First Moment T1 | | | Second Moment T2 | | |
|---|---|---|---|---|---|---|
| Age* | Frequency | % | M (SD) | Frequency | % | M (SD) |
| Fifth | 201 | 11.6 | | 258 | 14.9 | |
| Sixth | 192 | 11.1 | | 215 | 12.4 | |
| Seventh | 93 | 5.4 | | 165 | 9.5 | |
| Eighth | 106 | 6.1 | | 115 | 6.6 | |
| Ninth | 23 | 1.3 | | 81 | 4.7 | |
| Tenth | 20 | 1.2 | | 38 | 2.2 | |
| Rotating internship | 6 | .3 | | 6 | .3 | |
| Average* | | | | | | |
| Excellent (9–10) | 315 | 18.2 | 8.28 (0.71) | 187 | 10.8 | 8.12 (0.72) |
| Very good (8–8.9) | 893 | 51.5 | | 893 | 51.5 | |
| Well (7–7.9) | 471 | 27.1 | | 546 | 31.5 | |
| Fail (< 7) | 56 | 3.2 | | 109 | 6.3 | |

*Variables are presented in the table in classes with intervals for easy visualization. However, statistical analyses were performed using the original data collected as whole numbers (age) and numbers with decimals (average).

### Themes: Lifestyle modifications

During the virtual learning modality (Category 1), a sedentary lifestyle and isolation affected various aspects of students' lives, such as their physical and mental health, social relationships, and daily routines. Moreover, they experienced changes in their eating habits and stress levels. They learned to value important aspects of life and adapt to this new reality.

> *"During isolation, there were alterations, anxiety, [and] visual problems due to [using] the computer."* Nursing student, female, 21 years old, fourth level.

> *"What affected me the most was my way of learning, because [learning remotely] is more difficult; being behind a computer for hours and hours damaged my eyesight."* Medical student, male, 18 years old, first level.

> *"My daily routine has changed a lot compared to before the pandemic. My life has been more physically and socially active since I've been going to the university daily."* Physiotherapy student, female, 22 years old, fifth level.

As for Category 1, the students expressed modifications to their lifestyles, allowing us to identify more relevant codes: in 24.7% of students mentioned "socialization problems," with a higher prevalence among women (16.4%) than men (8.2%). Medical students reported it more frequently (7.3% of the overall sample). "Learning and concentration problems" was the second most mentioned code by the students (19% of the total sample), with a difference between women and men (13.3% for women vs. 4.7% for men). This code was especially prevalent among dental students (4.7% of the overall sample).

Statistically significant differences were observed in the codes of "mental health problems and stress" (p = 0.02) and "lack of physical activity and sedentary lifestyle" (p = 0.05) in relation to the students' academic program and sex.

Medical students reported the highest incidence of "mental health problems and stress" (3.3% of the total sample), with a greater frequency among women (9.3% for women vs. 4.2% for men), whereas nursing students more frequently reported a "lack of physical activity and sedentary lifestyle" (2.1% of the total sample), with a greater frequency among women (13.5% for women vs. 2.7% for men).

Upon returning to face-to-face instruction (Category 2), the students reported being better and more active; some adapted well, whereas others had difficulty. Although they are still afraid of getting sick, they carry out activities via direct contact and share things with friends. Some have experienced economic and emotional problems, and greater awareness has been raised about the importance of health and the opportunity to reflect on life.

*"I feel excited about the experience, but I am aware of the academic gaps I have; that scares me."* Clinical laboratory student, male, 20 years old, second level.

*"I had to readjust to going to and from school, living without my friends, and having my own responsibilities."* Dentistry student, female, 24 years old, fourth level.

*"I had to leave [behind] many plans and activities that I was doing because of isolation. Much of what I had planned was never done, but now I am getting back on the right course in my life."* Clinical Psychology student, male, 21 years old, second level.

The code "lifestyle change/adaptation" was reported by 29.5% of the students. This code was more common among women (21.7%) than men (7.8%), being particularly notable in medical (8.4% of the total sample) and clinical psychology (4.7%) students. "Increased stress" was the second most common code (13.9% of the total sample), with a higher prevalence in women (10.4% for women vs. 3.5% for men) and especially in the academic program of dentistry (10.7% for women vs. 3.3% for men) (see Table 2).

**Themes: Modification of academic activity**

In the virtual learning modality (Category 1), the main negative effects included stress, a lack of practice, and academic problems. In addition, there were difficulties in adapting to change, lower academic performance, and a lack of social and family ties.

*"[I have been affected in] a negative way since the virtual modality has certain aspects that do not allow professionals to be effectively trained."* Nursing student, male, 21 years old, first level.

*"They [my problems] have caused me to have to make an extra effort to learn [about] topics of interest in the career, and at the same time, [they have] increased pressure [nerves] when I give a presentation."* Medical student, female, 21 years old, fifth level.

*"[I feel a] little bad because as a person who was used to interacting with other people, [virtual learning] helped me relax."* Physiotherapy student, male, 21 years old, fourth level.

"Concentration/attention problems" was the most frequent code, mentioned by 26.3% of the students. This code showed a statistically significant difference between academic program and sex (p = 0.012), with a higher prevalence among women (20.2%) than men (6.1%). Medical students reported the highest incidence of this problem (7.1% of the total sample).

"Learning difficulties and low performance" was the second most common code in this category, affecting 19.3% of students, with no statistically significant differences; a higher prevalence was observed among women (13.8%) than men (5.5%).

There was an improvement in learning upon the return to face-to-face activities (Category 2), especially in practical learning; however, stress and low concentration were also experienced due to a lack of time and poor organization of one's schedule. Some students found it difficult to adapt to the face-to-face modality after having gone through the virtual modality, whereas others enjoyed greater socialization and physical activity.

**Table 2. Lifestyle modification, association (X2) academic program, sex (n = 1735).**

| Category 1: Social isolation, virtual learning modality. | Nursing (n = 223) | | | | | | Medicine (n = 430) | | | | | | Physiotherapy (n = 231) | | | | | |
|---|---|---|---|---|---|---|---|---|---|---|---|---|---|---|---|---|---|---|
| | Male | | Female | | Total | | Male | | Female | | Total | | Male | | Female | | Total | |
| Code | fi | % | fi | % | fi | % | fi | % | fi | % | fi | % | fi | % | fi | % | fi | % |
| Lack of physical activity and sedentary lifestyle | 6 | 2.7 | 30 | 13.5 | 36 | 2.1 | 31 | 7.2 | 43 | 10.0 | 74 | 4.3 | 10 | 4.3 | 28 | 12.1 | 38 | 2.2 |
| Socialization problems | 8 | 3.6 | 40 | 17.9 | 48 | 2.8 | 49 | 11.4 | 77 | 17.9 | 126 | 7.3 | 15 | 6.5 | 35 | 15.2 | 50 | 2.9 |
| Mental health problems and stress | 6 | 2.7 | 30 | 13.5 | 36 | 2.1 | 18 | 4.2 | 40 | 9.3 | 58 | 3.3 | 5 | 2.2 | 24 | 10.4 | 29 | 1.7 |
| Feeding problems | 1 | 0.4 | 8 | 3.6 | 9 | 0.5 | 5 | 1.2 | 21 | 4.9 | 26 | 1.5 | 2 | 0.9 | 5 | 2.2 | 7 | 0.4 |
| Learning and concentration problems | 10 | 4.5 | 32 | 14.3 | 42 | 2.4 | 21 | 4.9 | 55 | 12.8 | 76 | 4.4 | 10 | 4.3 | 37 | 16.0 | 47 | 2.7 |
| Changes in routine | 0 | 0.0 | 2 | 0.9 | 2 | 0.1 | 2 | 0.5 | 1 | 0.2 | 3 | 0.2 | 0 | 0.0 | 1 | 0.4 | 1 | 0.1 |
| Family/relationship problems | 2 | 0.9 | 4 | 1.8 | 6 | 0.3 | 0 | 0.0 | 7 | 1.6 | 7 | 0.4 | 2 | 0.9 | 4 | 1.7 | 6 | 0.3 |
| Economic problems | 0 | 0.0 | 5 | 2.2 | 5 | 0.3 | 1 | 0.2 | 5 | 1.2 | 6 | 0.3 | 5 | 2.2 | 8 | 3.5 | 13 | 0.7 |
| Other | 4 | 1.8 | 35 | 15.7 | 39 | 2.2 | 19 | 4.4 | 35 | 8.1 | 54 | 3.1 | 12 | 5.2 | 28 | 12.1 | 40 | 2.3 |
| Total academic program | 37 | 16.6 | 186 | 83.4 | 223 | 12.9 | 146 | 34.0 | 284 | 66.0 | 430 | 24.8 | 61 | 26.4 | 170 | 73.6 | 231 | 13.3 |
| Category 2: The return to face-to-face activities. | Nursing (n = 223) | | | | | | Medicine (n = 430) | | | | | | Physiotherapy (n = 231) | | | | | |
| | Male | | Female | | Total | | Male | | Female | | Total | | Male | | Female | | Total | |
| Code | fi | % | fi | % | fi | % | fi | % | fi | % | fi | % | fi | % | fi | % | fi | % |
| No. | 6 | 2.7 | 35 | 15.7 | 41 | 2.4 | 35 | 8.1 | 58 | 13.5 | 93 | 5.4 | 12 | 5.2 | 48 | 20.8 | 60 | 3.5 |
| Lifestyle change/Adaptation | 7 | 3.1 | 45 | 20.2 | 52 | 3.0 | 48 | 11.2 | 98 | 22.8 | 146 | 8.4 | 20 | 8.7 | 51 | 22.1 | 71 | 4.1 |
| Increased stress | 6 | 2.7 | 34 | 15.2 | 40 | 2.3 | 20 | 4.7 | 35 | 8.1 | 55 | 3.2 | 5 | 2.2 | 18 | 7.8 | 23 | 1.3 |
| Socialization problems | 1 | 0.4 | 14 | 6.3 | 15 | 0.9 | 2 | 0.5 | 15 | 3.5 | 17 | 1.0 | 2 | 0.9 | 8 | 3.5 | 10 | 0.6 |
| Changes in routine/schedule | 6 | 2.7 | 28 | 12.6 | 34 | 2.0 | 11 | 2.6 | 38 | 8.8 | 49 | 2.8 | 10 | 4.3 | 30 | 13.0 | 40 | 2.3 |
| Decreased physical activity | 3 | 1.3 | 2 | 0.9 | 5 | 0.3 | 5 | 1.2 | 10 | 2.3 | 15 | 0.9 | 2 | 0.9 | 4 | 1.7 | 6 | 0.3 |
| Feeding problems | 1 | 0.4 | 8 | 3.6 | 9 | 0.5 | 1 | 0.2 | 6 | 1.4 | 7 | 0.4 | 1 | 0.4 | 1 | 0.4 | 2 | 0.1 |
| Lack of responsibility/independence | 1 | 0.4 | 4 | 1.8 | 5 | 0.3 | 5 | 1.2 | 4 | 0.9 | 9 | 0.5 | 1 | 0.4 | 2 | 0.9 | 3 | 0.2 |
| Other | 6 | 2.7 | 16 | 7.2 | 22 | 1.3 | 19 | 4.4 | 20 | 4.7 | 39 | 2.2 | 8 | 3.5 | 8 | 3.5 | 16 | 0.9 |
| Total academic program | 37 | 16.6 | 186 | 83.4 | 223 | 12.9 | 146 | 34.0 | 284 | 66.0 | 430 | 24.8 | 61 | 26.4 | 170 | 73.6 | 231 | 13.3 |

The percentages in the academic programs reflect the proportion of students who mentioned each code in relation to the total number of participants in their respective careers by sex. The percentages of the total sample reflect the proportion in relation to the total sample.

Note: fi = absolute frequency; % = percentage; p = value of the significance level.

*"Yes, it has affected my academic performance a lot. I cannot go home and rest a bit, so I feel tired, and it is difficult for me to concentrate. Sleeping well also seems not to have been an option since I started classes."* Clinical laboratory student, female, 20 years old, third level.

*"By moving from virtual to face-to-face education, the time factor involved in moving from one place to another was affected."* Dentistry student, male, 19 years old, first level.

*"Academically, I understand the classes better, and I have less stress in regard to doing homework."* Clinical Psychology student, female, 20 years old, second level.

Upon returning to face-to-face activities, "increased stress and anxiety" was the most prevalent code, mentioned by 29.4% of the total sample. This phenomenon revealed a statistically significant difference between academic program and sex (p = 0.029), with a higher prevalence among women (22%) than men (7.4%). Dental students reported the highest incidence of this problem (9.9% of the total sample), followed by medical students (6.6%).

| Clinical laboratory (n = 204) | | | | | | Dentistry (n = 393) | | | | | | Clinical psychology (n = 254) | | | | | | Total code | | | | Total n = 1735 | | p |
|---|---|---|---|---|---|---|---|---|---|---|---|---|---|---|---|---|---|---|---|---|---|---|---|---|
| Male | | Female | | Total | | Male | | Female | | Total | | Male | | Female | | Total | | Male | | Female | | | | |
| fi | % | fi | % | fi | % | fi | % | fi | % | fi | % | fi | % | fi | % | fi | % | fi | % | fi | % | fi | % | |
| 12 | 5.9 | 17 | 8.3 | 29 | 1.7 | 20 | 5.1 | 31 | 7.9 | 51 | 2.9 | 9 | 3.5 | 30 | 11.8 | 39 | 2.2 | 88 | 5.1 | 179 | 10.3 | 267 | 15.4 | 0.05 |
| 17 | 8.3 | 30 | 14.7 | 47 | 2.7 | 35 | 8.9 | 56 | 14.2 | 91 | 5.2 | 19 | 7.5 | 47 | 18.5 | 66 | 3.8 | 143 | 8.2 | 285 | 16.4 | 428 | 24.7 | 0.08 |
| 0 | 0.0 | 27 | 13.2 | 27 | 1.6 | 12 | 3.1 | 41 | 10.4 | 53 | 3.1 | 3 | 1.2 | 25 | 9.8 | 28 | 1.6 | 44 | 2.5 | 187 | 10.8 | 231 | 13.3 | 0.02 |
| 3 | 1.5 | 7 | 3.4 | 10 | 0.6 | 2 | 0.5 | 12 | 3.1 | 14 | 0.8 | 0 | 0.0 | 12 | 4.7 | 12 | 0.7 | 13 | 0.7 | 65 | 3.7 | 78 | 4.5 | 0.44 |
| 11 | 5.4 | 30 | 14.7 | 41 | 2.4 | 21 | 5.3 | 60 | 15.3 | 81 | 4.7 | 8 | 3.1 | 34 | 13.4 | 42 | 2.4 | 81 | 4.7 | 248 | 14.3 | 329 | 19.0 | 0.91 |
| 1 | 0.5 | 1 | 0.5 | 2 | 0.1 | 1 | 0.3 | 3 | 0.8 | 4 | 0.2 | 3 | 1.2 | 5 | 2.0 | 8 | 0.5 | 7 | 0.4 | 13 | 0.7 | 20 | 1.2 | 0.65 |
| 1 | 0.5 | 7 | 3.4 | 8 | 0.5 | 5 | 1.3 | 14 | 3.6 | 19 | 1.1 | 1 | 0.4 | 4 | 1.6 | 5 | 0.3 | 11 | 0.6 | 40 | 2.3 | 51 | 2.9 | 0.62 |
| 0 | 0.0 | 6 | 2.9 | 6 | 0.3 | 5 | 1.3 | 5 | 1.3 | 10 | 0.6 | 4 | 1.6 | 8 | 3.1 | 12 | 0.7 | 15 | 0.9 | 37 | 2.1 | 52 | 3.0 | 0.17 |
| 5 | 2.5 | 29 | 14.2 | 34 | 2.0 | 24 | 6.1 | 46 | 11.7 | 70 | 4.0 | 10 | 3.9 | 32 | 12.6 | 42 | 2.4 | 74 | 4.3 | 205 | 11.8 | 279 | 16.1 | 0.03 |
| 50 | 24.5 | 154 | 75.5 | 204 | 11.8 | 125 | 31.8 | 268 | 68.2 | 393 | 22.7 | 57 | 22.4 | 197 | 77.6 | 254 | 14.6 | 476 | 27.4 | 1259 | 72.6 | 1735 | 100 | 0.00 |

| Clinical laboratory (n = 204) | | | | | | Dentistry (n = 393) | | | | | | Clinical psychology (n = 254) | | | | | | Total code | | | | Total n = 1735 | | p |
|---|---|---|---|---|---|---|---|---|---|---|---|---|---|---|---|---|---|---|---|---|---|---|---|---|
| Male | | Female | | Total | | Male | | Female | | Total | | Male | | Female | | Total | | Male | | Female | | | | |
| fi | % | fi | % | fi | % | fi | % | fi | % | fi | % | fi | % | fi | % | fi | % | fi | % | fi | % | fi | % | |
| 13 | 6.4 | 33 | 16.2 | 46 | 2.7 | 30 | 7.6 | 58 | 14.8 | 88 | 5.1 | 13 | 5.1 | 41 | 16.1 | 54 | 3.1 | 109 | 6.3 | 273 | 15.7 | 382 | 22.0 | 0.04 |
| 12 | 5.9 | 50 | 24.5 | 62 | 3.6 | 30 | 7.6 | 69 | 17.6 | 99 | 5.7 | 18 | 7.1 | 63 | 24.8 | 81 | 4.7 | 135 | 7.8 | 376 | 21.7 | 511 | 29.5 | 0.06 |
| 7 | 3.4 | 19 | 9.3 | 26 | 1.5 | 13 | 3.3 | 42 | 10.7 | 55 | 3.2 | 9 | 3.5 | 33 | 13.0 | 42 | 2.4 | 60 | 3.5 | 181 | 10.4 | 241 | 13.9 | 0.26 |
| 3 | 1.5 | 12 | 5.9 | 15 | 0.9 | 4 | 1.0 | 27 | 6.9 | 31 | 1.8 | 2 | 0.8 | 8 | 3.1 | 10 | 0.6 | 14 | 0.8 | 84 | 4.8 | 98 | 5.6 | 0.88 |
| 7 | 3.4 | 21 | 10.3 | 28 | 1.6 | 23 | 5.9 | 31 | 7.9 | 54 | 3.1 | 6 | 2.4 | 20 | 7.9 | 26 | 1.5 | 63 | 3.6 | 168 | 9.7 | 231 | 13.3 | 0.11 |
| 2 | 1.0 | 5 | 2.5 | 7 | 0.4 | 4 | 1.0 | 11 | 2.8 | 15 | 0.9 | 3 | 1.2 | 7 | 2.8 | 10 | 0.6 | 19 | 1.1 | 39 | 2.2 | 58 | 3.3 | 0.85 |
| 0 | 0.0 | 1 | 0.5 | 1 | 0.1 | 2 | 0.5 | 4 | 1.0 | 6 | 0.3 | 0 | 0.0 | 2 | 0.8 | 2 | 0.1 | 5 | 0.3 | 22 | 1.3 | 27 | 1.6 | 0.66 |
| 2 | 1.0 | 3 | 1.5 | 5 | 0.3 | 3 | 0.8 | 4 | 1.0 | 7 | 0.4 | 1 | 0.4 | 2 | 0.8 | 3 | 0.2 | 13 | 0.7 | 19 | 1.1 | 32 | 1.8 | 0.87 |
| 4 | 2.0 | 10 | 4.9 | 14 | 0.8 | 16 | 4.1 | 22 | 5.6 | 38 | 2.2 | 5 | 2.0 | 21 | 8.3 | 26 | 1.5 | 58 | 3.3 | 97 | 5.6 | 155 | 8.9 | 0.12 |
| 50 | 24.5 | 154 | 75.5 | 204 | 11.8 | 125 | 31.8 | 268 | 68.2 | 393 | 22.7 | 57 | 22.4 | 197 | 77.6 | 254 | 14.6 | 476 | 27.4 | 1259 | 72.6 | 1735 | 100 | 0.00 |

The second most frequent code was "adaptation difficulties," which was named by 16% of the students, with a statistically significant difference (p = 0.011) between academic program and sex. Medical students had the highest prevalence (4.4% of the total sample), with a higher incidence among women (10%) than men (7.7%) in terms of academic program.

The code "positive changes/none" was reported by 36.8% of the sample, suggesting that a significant proportion of the students did not experience negative changes or even perceived improvements during this period (see Table 3).

### Themes: Preference for vocational training modality

During social isolation, most students preferred the face-to-face learning modality and a return to face-to-face activities due to the importance of practice and interaction with teachers and classmates. However, some opted for the virtual modality for economic reasons or convenience. They also mentioned a combination of both modalities. In general, the choice depended on the needs and circumstances of each student.

Category 1: Mandatory social isolation, virtual learning modality

*"[I prefer] face-to-face because the interaction between the student and the teacher is more direct, and there are no distractions as there are at home."* Clinical Psychology student, male, 21 years old, sixth level.

Table 3. Modification of academic activity, association (X²). Academic program, sex (n = 1735).

| Category 1: Social isolation, virtual learning modality. | Nursing (n = 223) | | | | | | Medicine (n = 430) | | | | | | Physiotherapy (n = 231) | | | | | |
|---|---|---|---|---|---|---|---|---|---|---|---|---|---|---|---|---|---|---|
| | Male | | Female | | Total | | Male | | Female | | Total | | Male | | Female | | Total | |
| Code | fi | % | fi | % | fi | % | fi | % | fi | % | fi | % | fi | % | fi | % | fi | % |
| Stress generation/increase | 5 | 2.2 | 8 | 3.6 | 13 | 0.7 | 17 | 4.0 | 23 | 5.3 | 40 | 2.3 | 2 | 0.9 | 5 | 2.2 | 7 | 0.4 |
| Problems with concentration/attention | 6 | 2.7 | 61 | 27.4 | 67 | 3.9 | 38 | 8.8 | 86 | 20.0 | 124 | 7.1 | 11 | 4.8 | 37 | 16.0 | 48 | 2.8 |
| Learning difficulties and poor performance | 10 | 4.5 | 33 | 14.8 | 43 | 2.5 | 20 | 4.7 | 53 | 12.3 | 73 | 4.2 | 18 | 7.8 | 33 | 14.3 | 51 | 2.9 |
| Lack of practice/knowledge | 7 | 3.1 | 15 | 6.7 | 22 | 1.3 | 12 | 2.8 | 22 | 5.1 | 34 | 2.0 | 5 | 2.2 | 18 | 7.8 | 23 | 1.3 |
| Adjustment difficulties | 1 | 0.4 | 9 | 4.0 | 10 | 0.6 | 8 | 1.9 | 12 | 2.8 | 20 | 1.2 | 1 | 0.4 | 10 | 4.3 | 11 | 0.6 |
| Lack of motivation/tiredness | 4 | 1.8 | 15 | 6.7 | 19 | 1.1 | 16 | 3.7 | 19 | 4.4 | 35 | 2.0 | 5 | 2.2 | 17 | 7.4 | 22 | 1.3 |
| Other | 4 | 1.8 | 45 | 20.2 | 49 | 2.8 | 35 | 8.1 | 69 | 16.0 | 104 | 6.0 | 19 | 8.2 | 50 | 21.6 | 69 | 4.0 |
| Total | 37 | 16.6 | 186 | 83.4 | 223 | 12.9 | 146 | 34.0 | 284 | 66.0 | 430 | 24.8 | 61 | 26.4 | 170 | 73.6 | 231 | 13.3 |
| Category 2: The return to face-to-face activities. | Nursing (n = 223) | | | | | | Medicine (n = 430) | | | | | | Physiotherapy (n = 231) | | | | | |
| | Male | | Female | | Total | | Male | | Female | | Total | | Male | | Female | | Total | |
| Code | fi | % | fi | % | fi | % | fi | % | fi | % | fi | % | fi | % | fi | % | fi | % |
| Increased stress and anxiety | 8 | 3.6 | 52 | 23.3 | 60 | 3.5 | 30 | 7.0 | 85 | 19.8 | 115 | 6.6 | 11 | 4.8 | 36 | 15.6 | 47 | 2.7 |
| Learning difficulties/academic gaps | 5 | 2.2 | 19 | 8.5 | 24 | 1.4 | 20 | 4.7 | 33 | 7.7 | 53 | 3.1 | 4 | 1.7 | 19 | 8.2 | 23 | 1.3 |
| Adjustment difficulties | 3 | 1.3 | 17 | 7.6 | 20 | 1.2 | 33 | 7.7 | 43 | 10.0 | 76 | 4.4 | 9 | 3.9 | 20 | 8.7 | 29 | 1.7 |
| Difficulties in clinical practice | 7 | 3.1 | 9 | 4.0 | 16 | 0.9 | 0 | 0.0 | 1 | 0.2 | 1 | 0.1 | 5 | 2.2 | 8 | 3.5 | 13 | 0.7 |
| Low academic performance | 1 | 0.4 | 27 | 12.1 | 28 | 1.6 | 3 | 0.7 | 5 | 1.2 | 8 | 0.5 | 8 | 3.5 | 13 | 5.6 | 21 | 1.2 |
| Positive changes/None | 13 | 5.8 | 61 | 27.4 | 74 | 4.3 | 57 | 13.3 | 112 | 26.0 | 169 | 9.7 | 23 | 10.0 | 73 | 31.6 | 96 | 5.5 |
| Other | 0 | 0.0 | 1 | 0.4 | 1 | 0.1 | 3 | 0.7 | 5 | 1.2 | 8 | 0.5 | 1 | 0.4 | 1 | 0.4 | 2 | 0.1 |
| Total | 37 | 16.6 | 186 | 83.4 | 223 | 12.9 | 146 | 34.0 | 284 | 66.0 | 430 | 24.8 | 61 | 26.4 | 170 | 73.6 | 231 | 13.3 |

The percentages in the academic programs reflect the share of students who mentioned each code in relation to the total number of participants in their respective careers by sex. The percentages of the total sample reflect the proportion in relation to the total sample.

Note: fi = absolute frequency; % = percentage; p = value of the significance level.

*"[I prefer] face-to-face since it allows us to improve our knowledge and skills when doing internships, which is a fundamental part of our profession."* Dentistry student, female, 33 years old, eighth level.

*"Blended learning: the theory is very important to record on video and see as many times as possible when the subject and practice are not understood if 100% face-to-face."* Clinical laboratory student, male, 21 years old, fifth level.

*"[I prefer] virtual because I feel that studying depends on oneself and I consider myself responsible to be the protagonist of my study."* Physiotherapy student, female, 18 years old, first level.

During social isolation and the virtual learning modality (Category 1), students preferred the face-to-face modality. The code "face-to-face: facilitates learning" was the most common, mentioned by 27.1% of the students. Medical students had the highest incidence of this code (6.5% of the total sample). The second most common code was "face-to-face: the importance of internships/practical activities," mentioned by 13.6% of the students. This code showed a statistically significant difference between careers and sex (p = 0.023), with a higher prevalence among women (9%) than men (4.6%). Medical and dental students reported the highest incidence of this code (3.8% and 3.7% of the total sample, respectively).

The code "face-to-face: interaction/socialization problems" also showed a statistically significant difference between academic programs and sex (p = 0.015), reported by 8.6% of the students. This suggests that interaction and socialization challenges in returning to face-to-face activities varied significantly between the groups.

| Clinical laboratory (n = 204) | | | | | | Dentistry (n = 393) | | | | | | Clinical psychology (n = 254) | | | | | | Total code | | | | Total | | p |
|---|---|---|---|---|---|---|---|---|---|---|---|---|---|---|---|---|---|---|---|---|---|---|---|---|
| Male | | Female | | Total | | Male | | Female | | Total | | Male | | Female | | Total | | Male | | Female | | n = 1735 | | |
| fi | % | fi | % | fi | % | fi | % | fi | % | fi | % | fi | % | fi | % | fi | % | fi | % | fi | % | fi | % | |
| 3 | 1.5 | 15 | 7.4 | 18 | 1.0 | 6 | 1.5 | 25 | 6.4 | 31 | 1.8 | 5 | 2.0 | 15 | 5.9 | 20 | 1.2 | 38 | 2.2 | 91 | 5.2 | 129 | 7.4 | 0.227 |
| 10 | 4.9 | 39 | 19.1 | 49 | 2.8 | 29 | 7.4 | 74 | 18.8 | 103 | 5.9 | 11 | 4.3 | 54 | 21.3 | 65 | 3.7 | 105 | 6.1 | 351 | 20.2 | 456 | 26.3 | 0.012 |
| 10 | 4.9 | 23 | 11.3 | 33 | 1.9 | 24 | 6.1 | 53 | 13.5 | 77 | 4.4 | 13 | 5.1 | 45 | 17.7 | 58 | 3.3 | 95 | 5.5 | 240 | 13.8 | 335 | 19.3 | 0.675 |
| 5 | 2.5 | 15 | 7.4 | 20 | 1.2 | 10 | 2.5 | 33 | 8.4 | 43 | 2.5 | 3 | 1.2 | 10 | 3.9 | 13 | 0.7 | 42 | 2.4 | 113 | 6.5 | 155 | 8.9 | 0.819 |
| 5 | 2.5 | 9 | 4.4 | 14 | 0.8 | 5 | 1.3 | 14 | 3.6 | 19 | 1.1 | 3 | 1.2 | 8 | 3.1 | 11 | 0.6 | 23 | 1.3 | 62 | 3.6 | 85 | 4.9 | 0.357 |
| 5 | 2.5 | 10 | 4.9 | 15 | 0.9 | 4 | 1.0 | 10 | 2.5 | 14 | 0.8 | 7 | 2.8 | 19 | 7.5 | 26 | 1.5 | 41 | 2.4 | 90 | 5.2 | 131 | 7.6 | 0.372 |
| 12 | 5.9 | 43 | 21.1 | 55 | 3.2 | 47 | 12.0 | 59 | 15.0 | 106 | 6.1 | 15 | 5.9 | 46 | 18.1 | 61 | 3.5 | 132 | 7.6 | 312 | 18.0 | 444 | 25.6 | 0.000 |
| 50 | 24.5 | 154 | 75.5 | 204 | 11.8 | 125 | 31.8 | 268 | 68.2 | 393 | 22.7 | 57 | 22.4 | 197 | 77.6 | 254 | 14.6 | 476 | 27.4 | 1259 | 72.6 | 1735 | 100 | 0.000 |

| Clinical Laboratory (n = 204) | | | | | | Dentistry (n = 393) | | | | | | Clinical psychology (n = 254) | | | | | | Total code | | | | Total | | p |
|---|---|---|---|---|---|---|---|---|---|---|---|---|---|---|---|---|---|---|---|---|---|---|---|---|
| Male | | Female | | Total | | Male | | Female | | Total | | Male | | Female | | Total | | Male | | Female | | n = 1735 | | |
| fi | % | fi | % | fi | % | fi | % | fi | % | fi | % | fi | % | fi | % | fi | % | fi | % | fi | % | fi | % | |
| 11 | 5.4 | 45 | 22.1 | 56 | 3.2 | 57 | 14.5 | 114 | 29.0 | 171 | 9.9 | 12 | 4.7 | 49 | 19.3 | 61 | 3.5 | 129 | 7.4 | 381 | 22.0 | 510 | 29.39 | 0.029 |
| 5 | 2.5 | 21 | 10.3 | 26 | 1.5 | 8 | 2.0 | 13 | 3.3 | 21 | 1.2 | 4 | 1.6 | 13 | 5.1 | 17 | 1.0 | 46 | 2.7 | 118 | 6.8 | 164 | 9.4 | 0.252 |
| 9 | 4.4 | 31 | 15.2 | 40 | 2.3 | 15 | 3.8 | 33 | 8.4 | 48 | 2.8 | 11 | 4.3 | 53 | 20.9 | 64 | 3.7 | 80 | 4.6 | 197 | 11.4 | 277 | 15.97 | 0.011 |
| 0 | 0.0 | 1 | 0.5 | 1 | 0.1 | 6 | 1.5 | 4 | 1.0 | 10 | 0.6 | 0 | 0.0 | 0 | 0.0 | 0 | 0.0 | 18 | 1.0 | 23 | 1.3 | 41 | 2.3 | 0.596 |
| 0 | 0.0 | 4 | 2.0 | 4 | 0.2 | 1 | 0.3 | 5 | 1.3 | 6 | 0.3 | 0 | 0.0 | 5 | 2.0 | 5 | 0.3 | 13 | 0.7 | 59 | 3.4 | 72 | 4.15 | 0.018 |
| 23 | 11.3 | 49 | 24.0 | 72 | 4.1 | 35 | 8.9 | 90 | 22.9 | 125 | 7.2 | 28 | 11.0 | 75 | 29.5 | 103 | 5.9 | 179 | 10.3 | 460 | 26.5 | 639 | 36.83 | 0.150 |
| 2 | 1.0 | 3 | 1.5 | 5 | 0.3 | 3 | 0.8 | 9 | 2.3 | 12 | 0.7 | 2 | 0.8 | 2 | 0.8 | 4 | 0.2 | 11 | 0.6 | 21 | 1.2 | 32 | 1.844 | 0.883 |
| 50 | 24.5 | 154 | 75.5 | 204 | 11.8 | 125 | 31.8 | 268 | 68.2 | 393 | 22.7 | 57 | 22.4 | 197 | 77.6 | 254 | 14.6 | 476 | 27.4 | 1259 | 72.6 | 1735 | 100 | 0.000 |

## Category 2: The return to face-to-face activities

*"[I prefer] face-to-face because in that way there is already greater commitment, better interrelation between colleagues, and at the same time, traditional teaching will never be replaced by virtual education."* Medical student, male, 24 years old, seventh level.

*"[I prefer] face-to-face because it is more dynamic, and if we have any doubts, I know that the teacher will be there to explain it to us."* Nursing student, female, 25 years old, fifth level.

*"[I prefer] face-to-face because, although it is more difficult, you learn in a better way and in practice, you apply the theory, which is fundamental."* Dentistry student, male, 20 years old, second level.

*"[I prefer] hybrid because there are subjects that are theoretical and can be taught virtually while practice cannot."* Physiotherapy student, female, 23 years old, sixth level.

In Category 2, students' preference for the face-to-face modality was maintained. The code "face-to-face: facilitates learning" was the most common, mentioned by 22.9% of the total sample, with a higher prevalence among women (16.8%) than men (6.1%). Although no statistically significant differences were observed between academic program and sex, medical students reported the highest incidence of this code (4.9% of the total sample).

The second most frequent code was "face-to-face: the importance of internships/practical activities," mentioned by 14.3% of the students, with a higher prevalence among women (10.5%) than men (3.7%). Although there were no

**Table 4. Learning modality, association (X2) academic program, sex (n = 1735).**

| Category 1: Social isolation, virtual learning modality. | Nursing (n = 223) | | | | | | Medicine (n = 430) | | | | | | Physiotherapy (n = 231) | | | | | |
|---|---|---|---|---|---|---|---|---|---|---|---|---|---|---|---|---|---|---|
| | Male | | Female | | Total | | Male | | Female | | Total | | Male | | Female | | Total | |
| Code | fi | % | fi | % | fi | % | fi | % | fi | % | fi | % | fi | % | fi | % | fi | % |
| Face-to-face: The importance of internships/practical activities | 8 | 3.6 | 27 | 12.1 | 35 | 2.0 | 31 | 7.2 | 35 | 8.1 | 66 | 3.8 | 9 | 3.9 | 20 | 8.7 | 29 | 1.7 |
| Face-to-face: Facilitates learning | 9 | 4 | 42 | 18.8 | 51 | 2.9 | 36 | 8.4 | 77 | 17.9 | 113 | 6.5 | 17 | 7.4 | 63 | 27.3 | 80 | 4.6 |
| Face-to-face: Interaction/socialization problems | 1 | 0.4 | 12 | 5.4 | 13 | 0.7 | 18 | 4.2 | 18 | 4.2 | 36 | 2.1 | 6 | 2.6 | 10 | 4.3 | 16 | 0.9 |
| Face-to-face: Lack of dynamism | 0 | 0 | 3 | 1.3 | 3 | 0.2 | 1 | 0.2 | 4 | 0.9 | 5 | 0.3 | 0 | 0.0 | 2 | 0.9 | 2 | 0.1 |
| Face-to-face: No argument | 13 | 5.8 | 77 | 34.5 | 90 | 5.2 | 53 | 12.3 | 127 | 29.5 | 180 | 10.4 | 20 | 8.7 | 48 | 20.8 | 68 | 3.9 |
| Virtual: Economic hardship | 0 | 0 | 2 | 0.9 | 2 | 0.1 | 0 | 0 | 0 | 0 | 0 | 0.0 | 0 | 0.0 | 1 | 0.4 | 1 | 0.1 |
| Virtual: Family/Health relationship | 0 | 0 | 3 | 1.3 | 3 | 0.2 | 0 | 0 | 2 | 0.5 | 2 | 0.1 | 1 | 0.4 | 5 | 2.2 | 6 | 0.3 |
| Virtual: Self-study | 0 | 0 | 3 | 1.3 | 3 | 0.2 | 1 | 0.2 | 1 | 0.2 | 2 | 0.1 | 3 | 1.3 | 8 | 3.5 | 11 | 0.6 |
| Virtual: No plot | 4 | 1.8 | 7 | 3.1 | 11 | 0.6 | 3 | 0.7 | 9 | 2.1 | 12 | 0.7 | 2 | 0.9 | 8 | 3.5 | 10 | 0.6 |
| Hybrid: Ease of education | 1 | 0.4 | 4 | 1.8 | 5 | 0.3 | 1 | 0.2 | 6 | 1.4 | 7 | 0.4 | 2 | 0.9 | 3 | 1.3 | 5 | 0.3 |
| Hybrid: Economic/employment | 0 | 0 | 1 | 0.4 | 1 | 0.1 | 0 | 0 | 0 | 0 | 0 | 0 | 0 | 0 | 0 | 0.0 | 0 | 0 |
| Hybrid: Does not argue | 1 | 0.4 | 5 | 2.2 | 6 | 0.3 | 2 | 0.5 | 5 | 1.2 | 7 | 0.4 | 1 | 0.4 | 2 | 0.9 | 3 | 0.2 |
| Total | 37 | 16.6 | 186 | 83.4 | 223 | 12.9 | 146 | 34 | 284 | 66 | 430 | 24.8 | 61 | 26.4 | 170 | 73.6 | 231 | 13.3 |
| Category 2: The return to face-to-face activities. | Nursing (n = 223) | | | | | | Medicine (n = 430) | | | | | | Physiotherapy (n = 231) | | | | | |
| | Male | | Female | | Total | | Male | | Female | | Total | | Male | | Female | | Total | |
| Code | fi | % | fi | % | fi | % | fi | % | fi | % | fi | % | fi | % | fi | % | fi | % |
| Face-to-face: The importance of internships/practical activities | 3 | 1.3 | 21 | 9.4 | 24 | 1.4 | 21 | 4.9 | 45 | 10.5 | 66 | 3.8 | 9 | 3.9 | 28 | 12.1 | 37 | 2.1 |
| Face-to-face: Facilitates learning | 8 | 3.6 | 38 | 17.0 | 46 | 2.7 | 24 | 5.6 | 61 | 14.2 | 85 | 4.9 | 20 | 8.7 | 47 | 20.3 | 67 | 3.9 |
| Face-to-face: Interaction/socialization | 1 | 0.4 | 10 | 4.5 | 11 | 0.6 | 16 | 3.7 | 21 | 4.9 | 37 | 2.1 | 5 | 2.2 | 11 | 4.8 | 16 | 0.9 |
| Face-to-face: No argument | 21 | 9.4 | 90 | 40.4 | 111 | 6.4 | 74 | 17.2 | 122 | 28.4 | 196 | 11.3 | 17 | 7.4 | 60 | 26.0 | 77 | 4.4 |
| Virtual: Family aspect/health | 0 | 0.0 | 1 | 0.4 | 1 | 0.1 | 0 | 0.0 | 3 | 0.7 | 3 | 0.2 | 3 | 1.3 | 3 | 1.3 | 6 | 0.3 |
| Virtual: Economy | 0 | 0.0 | 0 | 0.0 | 0 | 0.0 | 0 | 0.0 | 3 | 0.7 | 3 | 0.2 | 0 | 0.0 | 3 | 1.3 | 3 | 0.2 |
| Virtual: Quality of learning | 0 | 0.0 | 0 | 0.0 | 0 | 0.0 | 1 | 0.2 | 4 | 0.9 | 5 | 0.3 | 0 | 0.0 | 2 | 0.9 | 2 | 0.1 |
| Virtual: No plot | 2 | 0.9 | 14 | 6.3 | 16 | 0.9 | 5 | 1.2 | 6 | 1.4 | 11 | 0.6 | 4 | 1.7 | 12 | 5.2 | 16 | 0.9 |
| Hybrid: Family/Health. | 0 | 0.0 | 0 | 0.0 | 0 | 0.0 | 0 | 0.0 | 3 | 0.7 | 3 | 0.2 | 1 | 0.4 | 1 | 0.4 | 2 | 0.1 |
| Hybrid: Quality learning and time | 1 | 0.4 | 4 | 1.8 | 5 | 0.3 | 0 | 0.0 | 4 | 0.9 | 4 | 0.2 | 0 | 0.0 | 1 | 0.4 | 1 | 0.1 |
| Hybrid: Does not argue | 1 | 0.4 | 8 | 3.6 | 9 | 0.5 | 5 | 1.2 | 12 | 2.8 | 17 | 1.0 | 2 | 0.9 | 2 | 0.9 | 4 | 0.2 |
| Total | 37 | 16.6 | 186 | 83.4 | 223 | 12.9 | 146 | 34.0 | 284 | 66.0 | 430 | 24.8 | 61 | 26.4 | 170 | 73.6 | 231 | 13.3 |

The percentages in the academic programs reflect the share of students who mentioned each code in relation to the total number of participants in their respective careers by sex. The percentages of the total sample reflect the proportion in relation to the total sample.

Note: fi = absolute frequency; % = percentage; p = value of the significance level.

statistically significant differences, dental students presented the highest incidence of this code (3.9% of the total sample). The code "face-to-face: no argument" was reported by 41% of the sample, with a statistically significant difference between academic program and sex (p = 0.001) (see Table 4).

## Themes: Academic stress

Confinement and virtual education during the COVID-19 pandemic (Category 1) caused students to experience high levels of anxiety and depression. Furthermore, they lacked social interaction, were overloaded by tasks, had technical problems, and had difficulty adapting to the new study system. They also experienced connectivity issues, a lack of resources,

| Clinical laboratory (n = 204) | | | | | | Dentistry (n = 393) | | | | | | Clinical psychology (n = 254) | | | | | | Total code | | | | Total | | p |
|---|---|---|---|---|---|---|---|---|---|---|---|---|---|---|---|---|---|---|---|---|---|---|---|---|
| Male | | Female | | Total | | Male | | Female | | Total | | Male | | Female | | Total | | Male | | Female | | n = 1735 | | |
| fi | % | fi | % | fi | % | fi | % | fi | % | fi | % | fi | % | fi | % | fi | % | fi | % | fi | % | fi | % | |
| 5 | 2.5 | 29 | 14.2 | 34 | 2 | 24 | 6.1 | 40 | 10.2 | 64 | 3.7 | 3 | 1.2 | 5 | 2.0 | 8 | 0.5 | 80 | 4.6 | 156 | 9.0 | 236 | 13.6 | 0,023 |
| 17 | 8.3 | 44 | 21.6 | 61 | 3.5 | 28 | 7.1 | 54 | 13.7 | 82 | 4.7 | 23 | 9.1 | 61 | 24.0 | 84 | 4.8 | 130 | 27.6 | 341 | 72.4 | 471 | 27.1 | 0,226 |
| 6 | 2.9 | 14 | 6.9 | 20 | 1.2 | 8 | 2.0 | 19 | 4.8 | 27 | 1.6 | 6 | 2.4 | 32 | 12.6 | 38 | 2.2 | 45 | 30.0 | 105 | 70.0 | 150 | 8.6 | 0,015 |
| 1 | 0.5 | 1 | 0.5 | 2 | 0.1 | 2 | 0.5 | 2 | 0.5 | 4 | 0.2 | 1 | 0.4 | 3 | 1.2 | 4 | 0.2 | 5 | 25.0 | 15 | 75.0 | 20 | 1.2 | 0,588 |
| 13 | 6.4 | 29 | 14.2 | 42 | 2.4 | 43 | 10.9 | 95 | 24.2 | 138 | 8.0 | 18 | 7.1 | 53 | 20.9 | 71 | 4.1 | 160 | 27.2 | 429 | 72.8 | 589 | 33.9 | 0,089 |
| 0 | 0.0 | 7 | 3.4 | 7 | 0.4 | 0 | 0.0 | 5 | 1.3 | 5 | 0.3 | 1 | 0.4 | 4 | 1.6 | 5 | 0.3 | 1 | 5.0 | 19 | 95.0 | 20 | 1.2 | 0,532 |
| 0 | 0.0 | 2 | 1.0 | 2 | 0.1 | 0 | 0.0 | 3 | 0.8 | 3 | 0.2 | 1 | 0.4 | 7 | 2.8 | 8 | 0.5 | 2 | 8.3 | 22 | 91.7 | 24 | 1.4 | 0,897 |
| 1 | 0.5 | 6 | 2.9 | 7 | 0.4 | 4 | 1.0 | 4 | 1.0 | 8 | 0.5 | 0 | 0.0 | 8 | 3.1 | 8 | 0.5 | 9 | 23.1 | 30 | 76.9 | 39 | 2.2 | 0,168 |
| 2 | 1.0 | 9 | 4.4 | 11 | 0.6 | 7 | 1.8 | 19 | 4.8 | 26 | 1.5 | 1 | 0.4 | 13 | 5.1 | 14 | 0.8 | 19 | 22.6 | 65 | 77.4 | 84 | 4.8 | 0,611 |
| 3 | 1.5 | 4 | 2.0 | 7 | 0.4 | 2 | 0.5 | 13 | 3.3 | 15 | 0.9 | 2 | 0.8 | 7 | 2.8 | 9 | 0.5 | 11 | 22.9 | 37 | 77.1 | 48 | 2.8 | 0,623 |
| 1 | 0.5 | 1 | 0.5 | 2 | 0.1 | 1 | 0.3 | 3 | 0.8 | 4 | 0.2 | 0 | 0.0 | 0 | 0.0 | 0 | 0.0 | 2 | 28.6 | 5 | 71.4 | 7 | 0.4 | 0,646 |
| 1 | 0.5 | 8 | 3.9 | 9 | 0.5 | 6 | 1.5 | 11 | 2.8 | 17 | 1.0 | 1 | 0.4 | 4 | 1.6 | 5 | 0.3 | 12 | 25.5 | 35 | 74.5 | 47 | 2.7 | 0,807 |
| 50 | 24.5 | 154 | 75.5 | 204 | 11.8 | 125 | 31.8 | 268 | 68.2 | 393 | 22.7 | 57 | 22.4 | 197 | 77.6 | 254 | 14.6 | 476 | 27.4 | 1259 | 72.6 | 1735 | 100.0 | 0,000 |

| Clinical laboratory (n = 204) | | | | | | Dentistry (n = 393) | | | | | | Clinical psychology (n = 254) | | | | | | Total code | | | | Total | | p |
|---|---|---|---|---|---|---|---|---|---|---|---|---|---|---|---|---|---|---|---|---|---|---|---|---|
| Male | | Female | | Total | | Male | | Female | | Total | | Male | | Female | | Total | | Male | | Female | | n = 1735 | | |
| fi | % | fi | % | fi | % | fi | % | fi | % | fi | % | fi | % | fi | % | fi | % | fi | % | fi | % | fi | % | |
| 8 | 3.9 | 35 | 17.2 | 43 | 2.5 | 23 | 5.9 | 45 | 11.5 | 68 | 3.92 | 1 | 0.4 | 9 | 3.5 | 10 | 0.6 | 65 | 3.7 | 183 | 10.5 | 248 | 14.3 | 0.148 |
| 14 | 6.9 | 36 | 17.6 | 50 | 2.9 | 19 | 4.8 | 46 | 11.7 | 65 | 3.75 | 20 | 7.9 | 64 | 25.2 | 84 | 4.8 | 105 | 6.1 | 292 | 16.8 | 397 | 22.9 | 0.685 |
| 0 | 0.0 | 2 | 1.0 | 2 | 0.1 | 8 | 2.0 | 14 | 3.6 | 22 | 1.27 | 6 | 2.4 | 25 | 9.8 | 31 | 1.8 | 36 | 2.1 | 83 | 4.8 | 119 | 6.9 | 0.140 |
| 19 | 9.3 | 60 | 29.4 | 79 | 4.6 | 56 | 14.2 | 105 | 26.7 | 161 | 9.28 | 19 | 7.5 | 68 | 26.8 | 87 | 5.0 | 206 | 11.9 | 505 | 29.1 | 711 | 41.0 | 0.001 |
| 0 | 0.0 | 2 | 1.0 | 2 | 0.1 | 0 | 0.0 | 2 | 0.5 | 2 | 0.12 | 0 | 0.0 | 3 | 1.2 | 3 | 0.2 | 3 | 0.2 | 14 | 0.8 | 17 | 1.0 | 0.246 |
| 0 | 0.0 | 2 | 1.0 | 2 | 0.1 | 0 | 0.0 | 2 | 0.5 | 2 | 0.12 | 0 | 0.0 | 2 | 0.8 | 2 | 0.1 | 0 | 0.0 | 12 | 0.7 | 12 | 0.7 | 0.000 |
| 0 | 0.0 | 1 | 0.5 | 1 | 0.1 | 1 | 0.3 | 2 | 0.5 | 3 | 0.17 | 1 | 0.4 | 4 | 1.6 | 5 | 0.3 | 3 | 0.2 | 13 | 0.7 | 16 | 0.9 | 0.891 |
| 3 | 1.5 | 10 | 4.9 | 13 | 0.7 | 6 | 1.5 | 18 | 4.6 | 24 | 1.38 | 3 | 1.2 | 14 | 5.5 | 17 | 1.0 | 23 | 1.3 | 74 | 4.3 | 97 | 5.6 | 0.497 |
| 1 | 0.5 | 5 | 2.5 | 6 | 0.3 | 3 | 0.8 | 6 | 1.5 | 9 | 0.52 | 1 | 0.4 | 2 | 0.8 | 3 | 0.2 | 6 | 0.3 | 17 | 1.0 | 23 | 1.3 | 0.689 |
| 0 | 0.0 | 1 | 0.5 | 1 | 0.1 | 2 | 0.5 | 6 | 1.5 | 8 | 0.46 | 3 | 1.2 | 2 | 0.8 | 5 | 0.3 | 6 | 0.3 | 18 | 1.0 | 24 | 1.4 | 0.377 |
| 5 | 2.5 | 0 | 0.0 | 5 | 0.3 | 7 | 1.8 | 22 | 5.6 | 29 | 1.67 | 3 | 1.2 | 4 | 1.6 | 7 | 0.4 | 23 | 1.3 | 48 | 2.8 | 71 | 4.1 | 0.014 |
| 50 | 24.5 | 154 | 75.5 | 204 | 11.8 | 125 | 31.8 | 268 | 68.2 | 393 | 22.7 | 57 | 22.4 | 197 | 77.6 | 254 | 14.6 | 476 | 27.4 | 1259 | 72.6 | 1735 | 100.0 | 0.000 |

and concerns about grades. Overall, stress and academic overload were among the main challenges experienced during this time.

*"I think it got to a point where my stress levels went up a lot without me realizing it. I was spending pretty much all day in my room doing computer homework, and I started to feel emotional and mental fatigue."* Clinical psychology student, female, 23 years old, seventh level.

*"It caused a lot of stress since for many of us, it was difficult to access the internet and have good equipment to take classes."* Dentistry student, male, 23 years old, eighth grade.

 

Table 5. Academic stress, association (X2), academic program, sex (n = 1735).

| Category 1: Social isolation. virtual learning modality. | Nursing (n = 223) | | | | | | Medicine (n = 430) | | | | | | Physiotherapy (n = 231) | | | | | |
|---|---|---|---|---|---|---|---|---|---|---|---|---|---|---|---|---|---|---|
| | Male | | Female | | Total | | Male | | Female | | Total | | Male | | Female | | Total | |
| Code | fi | % | fi | % | fi | % | fi | % | fi | % | fi | % | fi | % | fi | % | fi | % |
| No | 1 | 0.4 | 12 | 5.4 | 13 | 0.7 | 14 | 3.3 | 14 | 3.3 | 28 | 1.6 | 5 | 2.2 | 6 | 2.6 | 11 | 0.6 |
| Stress | 15 | 6.7 | 72 | 32.3 | 87 | 5.0 | 51 | 11.9 | 118 | 27.4 | 169 | 9.7 | 17 | 7.4 | 65 | 28.1 | 82 | 4.7 |
| Isolation/lack of socialization | 7 | 3.1 | 26 | 11.7 | 33 | 1.9 | 37 | 8.6 | 41 | 9.5 | 78 | 4.5 | 13 | 5.6 | 29 | 12.6 | 42 | 2.4 |
| Task overload | 1 | 0.4 | 9 | 4.0 | 10 | 0.6 | 1 | 0.2 | 14 | 3.3 | 15 | 0.9 | 6 | 2.6 | 4 | 1.7 | 10 | 0.6 |
| Lack of concentration | 0 | 0.0 | 4 | 1.8 | 4 | 0.2 | 2 | 0.5 | 11 | 2.6 | 13 | 0.7 | 1 | 0.4 | 2 | 0.9 | 3 | 0.2 |
| Learning disabilities | 8 | 3.6 | 29 | 13.0 | 37 | 2.1 | 23 | 5.3 | 39 | 9.1 | 62 | 3.6 | 9 | 3.9 | 32 | 13.9 | 41 | 2.4 |
| Lack of practice | 2 | 0.9 | 8 | 3.6 | 10 | 0.6 | 5 | 1.2 | 2 | 0.5 | 7 | 0.4 | 0 | 0.0 | 1 | 0.4 | 1 | 0.1 |
| Health problems | 1 | 0.4 | 12 | 5.4 | 13 | 0.7 | 5 | 1.2 | 12 | 2.8 | 17 | 1.0 | 5 | 2.2 | 12 | 5.2 | 17 | 1.0 |
| Connection issues | 0 | 0.0 | 2 | 0.9 | 2 | 0.1 | 0 | 0.0 | 4 | 0.9 | 4 | 0.2 | 0 | 0.0 | 5 | 2.2 | 5 | 0.3 |
| Other | 2 | 0.9 | 12 | 5.4 | 14 | 0.8 | 8 | 1.9 | 29 | 6.7 | 37 | 2.1 | 5 | 2.2 | 14 | 6.1 | 19 | 1.1 |
| Total | 37 | 16.6 | 186 | 83.4 | 223 | 12.9 | 146 | 34.0 | 284 | 66.0 | 430 | 24.8 | 61 | 26.4 | 170 | 73.6 | 231 | 13.3 |
| Category 2: The return to face-to-face activities. | Nursing (n = 223) | | | | | | Medicine (n = 430) | | | | | | Physiotherapy (n = 231) | | | | | |
| | Male | | Female | | Total | | Male | | Female | | Total | | Male | | Female | | Total | |
| Code | fi | % | fi | % | fi | % | fi | % | fi | % | fi | % | fi | % | fi | % | fi | % |
| No | 11 | 4.9 | 43 | 19.3 | 54 | 3.1 | 29 | 6.7 | 50 | 11.6 | 79 | 4.6 | 17 | 7.4 | 30 | 13.0 | 47 | 2.7 |
| Stress/anxiety | 7 | 3.1 | 86 | 38.6 | 93 | 5.4 | 66 | 15.3 | 144 | 33.5 | 210 | 12.1 | 29 | 12.6 | 80 | 34.6 | 109 | 6.3 |
| Difficulties in adapting to changes | 7 | 3.1 | 13 | 5.8 | 20 | 1.2 | 15 | 3.5 | 27 | 6.3 | 42 | 2.4 | 2 | 0.9 | 16 | 6.9 | 18 | 1.0 |
| Task overload | 4 | 1.8 | 19 | 8.5 | 23 | 1.3 | 6 | 1.4 | 19 | 4.4 | 25 | 1.4 | 1 | 0.4 | 12 | 5.2 | 13 | 0.7 |
| Schedule changes | 2 | 0.9 | 8 | 3.6 | 10 | 0.6 | 15 | 3.5 | 15 | 3.5 | 30 | 1.7 | 4 | 1.7 | 14 | 6.1 | 18 | 1.0 |
| Learning disabilities | 3 | 1.3 | 11 | 4.9 | 14 | 0.8 | 8 | 1.9 | 16 | 3.7 | 24 | 1.4 | 5 | 2.2 | 14 | 6.1 | 19 | 1.1 |
| Pressure/Demand | 0 | 0.0 | 3 | 1.3 | 3 | 0.2 | 4 | 0.9 | 9 | 2.1 | 13 | 0.7 | 2 | 0.9 | 1 | 0.4 | 3 | 0.2 |
| Economic hardship | 3 | 1.3 | 3 | 1.3 | 6 | 0.3 | 3 | 0.7 | 4 | 0.9 | 7 | 0.4 | 1 | 0.4 | 3 | 1.3 | 4 | 0.2 |
| Total | 37 | 16.6 | 186 | 83.4 | 223 | 12.9 | 146 | 34.0 | 284 | 66.0 | 430 | 24.8 | 61 | 26.4 | 170 | 73.6 | 231 | 13.3 |

The percentages in the academic programs reflect the share of students who mentioned each code in relation to the total number of participants in their respective careers by sex. The percentages of the total sample reflect the proportion in relation to the total sample.

Note: fi = absolute frequency; % = percentage; p = value of the significance level.

*"Social isolation, together with academic activities, has caused greater susceptibility to stress, so it is now much more common to reach very high levels of stress."* Clinical laboratory student, female, 20 years old, third level.

During social isolation and virtual learning, "stress" was the most frequent code, expressed by 37.4% of the students with no statistically significant differences. However, the prevalence in women (28%) was notably higher than that in men (9.5%).

"Learning disabilities" was the second most common code, affecting 16.3% of the total sample. Statistically significant differences were found in the codes "isolation/lack of socialization" (p = 0.016) and "task overload" (p = 0.011). Isolation affected 15.6% of the students, and task overload impacted 5.1%. These codes varied significantly between academic program and sex during virtual learning because of mandatory isolation.

The return to face-to-face activities (Category 2) has had various impacts on students. In some cases, increased stress is due to the academic load and difficulties in adapting to changes. However, there have also been opportunities for learning and academic improvement.

*"We reduce stress because in addition to improving our knowledge at university, we also interact again with our friends; therefore, we feel happier."* Physiotherapy student, male, 25 years old, eighth level.

| Clinical laboratory (n = 204) | | | | | | Dentistry (n = 393) | | | | | | Clinical psychology (n = 254) | | | | | | Total code | | | | Total n = 1735 | | p |
|---|---|---|---|---|---|---|---|---|---|---|---|---|---|---|---|---|---|---|---|---|---|---|---|---|
| Male | | Female | | Total | | Male | | Female | | Total | | Male | | Female | | Total | | Male | | Female | | | | |
| fi | % | fi | % | fi | % | fi | % | fi | % | fi | % | fi | % | fi | % | fi | % | fi | % | fi | % | fi | % | |
| 7 | 3.4 | 9 | 4.4 | 16 | 0.9 | 13 | 3.3 | 16 | 4.1 | 29 | 1.7 | 4 | 1.6 | 7 | 2.8 | 11 | 0.6 | 44 | 2.5 | 64 | 3.7 | 108 | 6.2 | 0.197 |
| 18 | 8.8 | 57 | 27.9 | 75 | 4.3 | 46 | 11.7 | 105 | 26.7 | 151 | 8.7 | 17 | 6.7 | 68 | 26.8 | 85 | 4.9 | 164 | 9.5 | 485 | 28.0 | 649 | 37.4 | 0.091 |
| 7 | 3.4 | 24 | 11.8 | 31 | 1.8 | 18 | 4.6 | 30 | 7.6 | 48 | 2.8 | 8 | 3.1 | 30 | 11.8 | 38 | 2.2 | 90 | 5.2 | 180 | 10.4 | 270 | 15.6 | 0.016 |
| 1 | 0.5 | 11 | 5.4 | 12 | 0.7 | 7 | 1.8 | 10 | 2.5 | 17 | 1.0 | 9 | 3.5 | 15 | 5.9 | 24 | 1.4 | 25 | 1.4 | 63 | 3.6 | 88 | 5.1 | 0.011 |
| 1 | 0.5 | 5 | 2.5 | 6 | 0.3 | 3 | 0.8 | 8 | 2.0 | 11 | 0.6 | 1 | 0.4 | 4 | 1.6 | 5 | 0.3 | 8 | 0.5 | 34 | 2.0 | 42 | 2.4 | 0.854 |
| 11 | 5.4 | 21 | 10.3 | 32 | 1.8 | 20 | 5.1 | 49 | 12.5 | 69 | 4.0 | 10 | 3.9 | 32 | 12.6 | 42 | 2.4 | 81 | 4.7 | 202 | 11.6 | 283 | 16.3 | 0.421 |
| 0 | 0.0 | 3 | 1.5 | 3 | 0.2 | 1 | 0.3 | 9 | 2.3 | 10 | 0.6 | 1 | 0.4 | 2 | 0.8 | 3 | 0.2 | 9 | 0.5 | 25 | 1.4 | 34 | 2.0 | 0.064 |
| 1 | 0.5 | 11 | 5.4 | 12 | 0.7 | 4 | 1.0 | 16 | 4.1 | 20 | 1.2 | 1 | 0.4 | 12 | 4.7 | 13 | 0.7 | 17 | 1.0 | 75 | 4.3 | 92 | 5.3 | 0.351 |
| 2 | 1.0 | 3 | 1.5 | 5 | 0.3 | 2 | 0.5 | 3 | 0.8 | 5 | 0.3 | 2 | 0.8 | 7 | 2.8 | 9 | 0.5 | 6 | 0.3 | 24 | 1.4 | 30 | 1.7 | 0.382 |
| 2 | 1.0 | 10 | 4.9 | 12 | 0.7 | 11 | 2.8 | 22 | 5.6 | 33 | 1.9 | 4 | 1.6 | 20 | 7.9 | 24 | 1.4 | 32 | 1.8 | 107 | 6.2 | 139 | 8.0 | 0.614 |
| 50 | 24.5 | 154 | 75.5 | 204 | 11.8 | 125 | 31.8 | 268 | 68.2 | 393 | 22.7 | 57 | 22.4 | 197 | 77.6 | 254 | 14.6 | 476 | 27.4 | 1259 | 72.6 | 1735 | 100 | 0.000 |

| Clinical laboratory (n = 204) | | | | | | Dentistry (n = 393) | | | | | | Clinical psychology (n = 254) | | | | | | Total code | | | | Total n = 1735 | | p |
|---|---|---|---|---|---|---|---|---|---|---|---|---|---|---|---|---|---|---|---|---|---|---|---|---|
| Male | | Female | | Total | | Male | | Female | | Total | | Male | | Female | | Total | | Male | | Female | | | | |
| fi | % | fi | % | fi | % | fi | % | fi | % | fi | % | fi | % | fi | % | fi | % | fi | % | fi | % | fi | % | |
| 12 | 5.9 | 34 | 16.7 | 46 | 2.7 | 33 | 8.4 | 47 | 12.0 | 80 | 4.6 | 12 | 4.7 | 33 | 13.0 | 45 | 2.6 | 114 | 6.6 | 237 | 13.7 | 351 | 20.2 | 0.113 |
| 21 | 10.3 | 67 | 32.8 | 88 | 5.1 | 46 | 11.7 | 124 | 31.6 | 170 | 9.8 | 18 | 7.1 | 103 | 40.6 | 121 | 7.0 | 187 | 10.8 | 604 | 34.8 | 791 | 45.6 | 0.000 |
| 9 | 4.4 | 14 | 6.9 | 23 | 1.3 | 14 | 3.6 | 26 | 6.6 | 40 | 2.3 | 9 | 3.5 | 22 | 8.7 | 31 | 1.8 | 56 | 3.2 | 118 | 6.8 | 174 | 10.0 | 0.445 |
| 5 | 2.5 | 14 | 6.9 | 19 | 1.1 | 7 | 1.8 | 18 | 4.6 | 25 | 1.4 | 8 | 3.1 | 11 | 4.3 | 19 | 1.1 | 31 | 1.8 | 93 | 5.4 | 124 | 7.1 | 0.316 |
| 1 | 0.5 | 8 | 3.9 | 9 | 0.5 | 5 | 1.3 | 12 | 3.1 | 17 | 1.0 | 2 | 0.8 | 15 | 5.9 | 17 | 1.0 | 29 | 1.7 | 72 | 4.1 | 101 | 5.8 | 0.049 |
| 1 | 0.5 | 14 | 6.9 | 15 | 0.9 | 18 | 4.6 | 25 | 6.4 | 43 | 2.5 | 7 | 2.8 | 9 | 3.5 | 16 | 0.9 | 42 | 2.4 | 89 | 5.1 | 131 | 7.6 | 0.137 |
| 0 | 0.0 | 1 | 0.5 | 1 | 0.1 | 1 | 0.3 | 7 | 1.8 | 8 | 0.5 | 1 | 0.4 | 2 | 0.8 | 3 | 0.2 | 8 | 0.5 | 23 | 1.3 | 31 | 1.8 | 0.416 |
| 1 | 0.5 | 2 | 1.0 | 3 | 0.2 | 1 | 0.3 | 9 | 2.3 | 10 | 0.6 | 0 | 0.0 | 2 | 0.8 | 2 | 0.1 | 9 | 0.5 | 23 | 1.3 | 32 | 1.8 | 0.461 |
| 61 | 29.9 | 170 | 83.3 | 231 | 13.3 | 125 | 31.8 | 268 | 68.2 | 393 | 22.7 | 57 | 22.4 | 197 | 77.6 | 254 | 14.6 | 476 | 27.4 | 1259 | 72.6 | 1735 | 100 | 0.000 |

*"My stress level declined due to my interpersonal and social relationships with my friends and teachers. I occasionally experienced increased stress due to lack of time resulting from the difficulty of my schedule and the amount of homework and studying."* Medical student, female, 20 years old, first level.

*"I have a heavy schedule. I hand in assignments in the virtual classroom before the next class. I do not have a good diet."* Nursing student, male, 22 years old, fourth level.

In Category 2, "stress/anxiety" was the most prevalent code, mentioned by 45.6% of the total sample. This code showed a statistically significant difference between academic program and sex (p < 0.001), with a higher prevalence among women (34.8%) than men (10.8%). Medical students reported the highest incidence of this problem (12.1% of the total sample), followed by dentistry students (9.8%).

"Difficulties in adapting to change" was the second most frequent code, affecting 10% of the students, with no statistically significant differences observed. A statistically significant difference was observed in the code "changes in schedules" (p = 0.049). affecting 5.8% of the students. These findings suggest that this stress factor varied significantly between the academic program and sex upon the return to face-to-face activities (see Table 5).

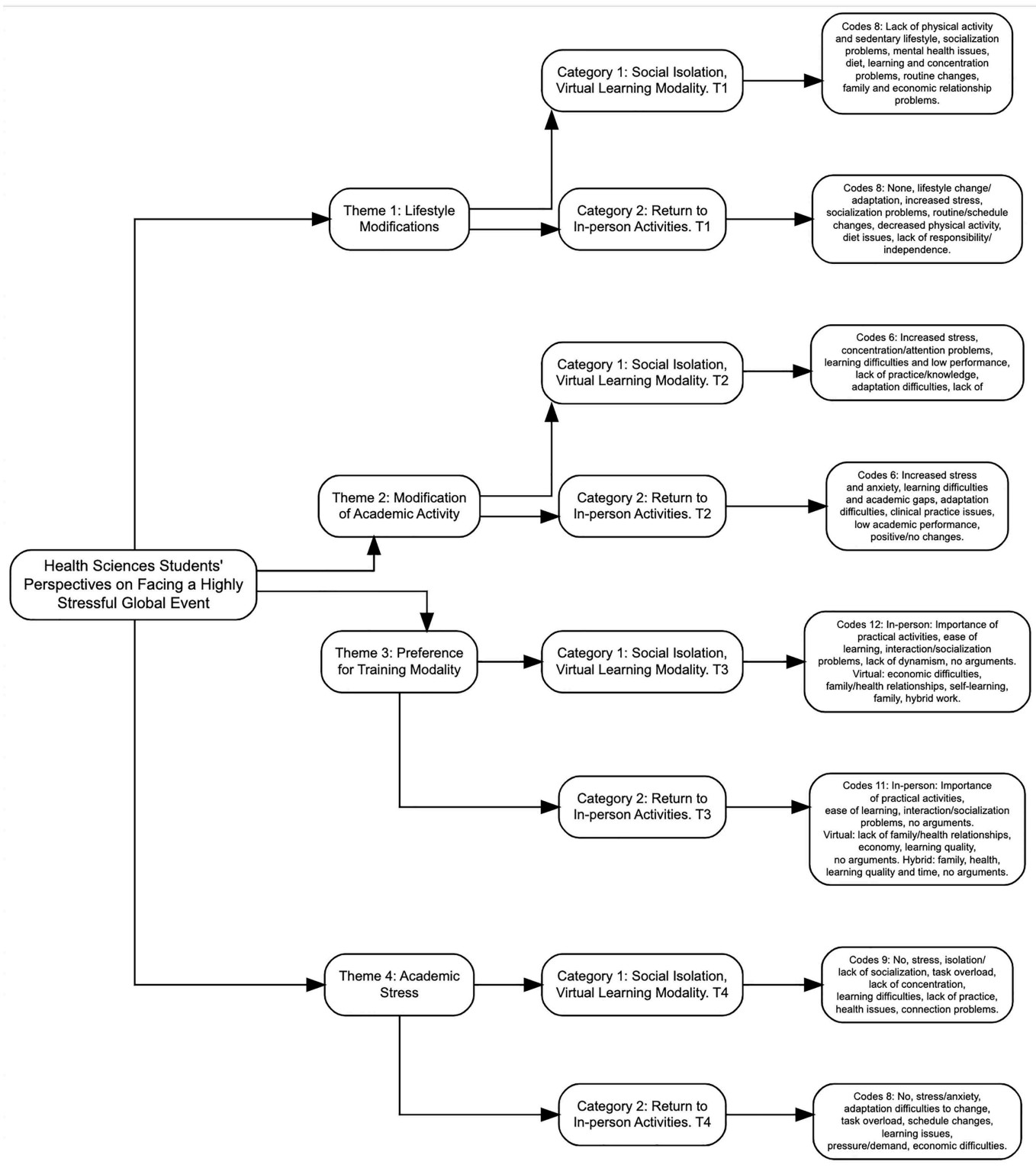

**Fig 1. General diagram of the results.**

## Discussion

For the present longitudinal qualitative study, the hermeneutic method was used to delve into the perspectives and experiences of the students in a particular context and time, the COVID-19 pandemic, widely recognized as a life-threatening global stressor [26–29], has particularly impacted university students through lifestyle restructuring and social limitations, affecting their mental health and subjective well-being by disrupting age-appropriate development and environmental conditions [30].

The inferential statistical analysis allowed us to examine the relationships between the variables of academic program and sex in at both time points. The findings revealed differences between male and female students in terms of the challenges faced during and after the COVID-19 pandemic, a highly stressful event.

The social limitations imposed to reduce the spread of the virus generated stressors such as the change to virtual education and social isolation, which caused a decrease in physical activity and sedentary lifestyle, eating problems these findings coincide with those of Gallé et al., who reported an increase in sedentary behavior and a decline in physical activity during this period [31]. Female students were affected by lifestyle changes and increased stress. This contributed to academic stress and a decrease in their interest in vocational training. These findings suggest that changes are required regarding the modality of education. As documented by Ruiz-Robledillo et al. [32], these factors significantly affect students' psychological and academic well-being. This is particularly the case in times of crisis, such as during the COVID-19 pandemic.

Higher education was severely disrupted due to restrictions imposed by governments to contain the spread of COVID-19. Social distancing measures [33] had a significant effect on university students, especially in the development of their academic activities, as documented by Bughrara et al. [34]. In addition, these dispositions modified students' lifestyle habits and mental health, as suggested by Michaeli et al. [35]. These approaches are in accordance with previous ones. Students at the University of Chimborazo perceived changes in their physical and mental health, as well as in their social relationships. Moreover, academic stress can occur due to changes in learning modality.

Physical activity, as part of students' lifestyles, decreased significantly during the COVID-19 pandemic and did not return to the usual pre-pandemic levels upon the return to face-to-face activities [36]. However, physical activity in the post-pandemic era was performed more frequently than during the pandemic. This is likely due to the decrease in social restrictions, an increase in the immunization rate, and greater access to public spaces such as gyms and parks [37]. The students of health sciences included in the present study showed greater post-pandemic activity but with certain restrictions due to their free time available, academic activities, and the risk of infection.

Online learning, as part of the isolation measures implemented during the COVID-19, highlights the digital divide among students, which is exacerbated by unfavorable economic and social situations. In addition to a sense of isolation and difficulties in adapting to online schedules and assignments, technical problems were a constant concern [38]. COVID-19 caused in students to experience inefficient learning and low participation as suggested by Zhu et al. [39]. Concerns were expressed about the lack of human contact, a sense of monotony, and the inadequacy of the information transferred as well as possible negative effects on physical health [40,41], with direct and indirect consequences that could influence students' quality as future health professionals [36].

In this context, the students of the Faculty of Health preferred the face-to-face modality during the COVID-19 pandemic and in the post-pandemic stage, considering the importance of clinical practice and direct interaction with professors and classmates.

The COVID-19 pandemic generated changes in the psychological health of the population, such as anxiety, loneliness, and, during the post-pandemic period, post-traumatic stress disorder [37,42]. These issues have not received adequate attention [4].

During confinement and the return to face-to-face classes, students in the university's six academic health programs experienced stress due to changes in the learning modality, task overload, and students' own difficulties in adapting to

changes. It is therefore necessary to implement infrastructure that allows for efficient provisions as well as sustainable, equitable mental health care [43].

As for the strengths, needs, and gaps in online learning, there is a need for human contact in health training, and technologies that support learning benefit training [44]. As stated by some students in this study, this motivated them to prefer the hybrid modality. This implies the design or redesign of curricula that combine learning modalities [40,45]. In addition, the digital competencies of teachers and students should be reinforced. This guarantees quality and support for learning in modalities other than face-to-face, with satisfactory outcomes.

### Limitations

There is a possibility of perception bias among participating students, derived from factors such as the epidemiological and cultural context. Individual beliefs may have influenced the students' opinions, which could limit the equitable representation of meaningful experiences.

### Recommendations

As for virtual education, the tools and online platforms used to ensure educational adaptations provide learning opportunities and flexibility for students. These adaptations will clearly remain part of education in the post-pandemic period. Hence, it is essential that educational institutions in the areas of health adopt and incorporate virtual education as an alternative to events that affect contexts temporarily into their curricula. In addition, it is necessary for institutions to prepare for future emergencies by teaching students effective coping and stress management strategies. This will ensure continuous and high-quality education for students in any situation.

### Conclusions

Remembering the lessons learned during the COVID-19 pandemic and overcoming the obstacles encountered are important; students understand the impact of lifestyle modifications and academic activities that lead to stress during emergencies. The students acquired new knowledge, thus guaranteeing effective and higher-quality lifestyle practices and stress management in the post-pandemic era.

The changes in students' physical and mental health during the COVID-19 pandemic were modified in the return to face-to-face activities. Students are sequentially recovering in terms of physical activity and social contact.

Learning virtually caused students to experience academic stress, they reflected on their clinical practice, academic performance, and lack of social and family contact upon. The return to face-to-face learning, the students experienced poor organization in terms of time and difficulty adapting to the face-to-face modality. However, they enjoyed greater interaction and social activities.

### Acknowledgments

We would like to express our thanks to the students who participated in the study.

### Author contributions

**Conceptualization:** Yolanda E. Salazar Granizo.

**Data curation:** Yolanda E. Salazar Granizo.

**Formal analysis:** Yolanda E. Salazar Granizo.

**Investigation:** Yolanda E. Salazar Granizo.

**Methodology:** Yolanda E. Salazar Granizo.

**Supervision:** Rafael A. Caparros-Gonzalez, Daniel Puente-Fernandez, César Hueso-Montoro.

**Validation:** Yolanda E. Salazar Granizo.

**Visualization:** Rafael A. Caparros-Gonzalez, Daniel Puente-Fernandez, César Hueso-Montoro.

**Writing – original draft:** Yolanda E. Salazar Granizo.

**Writing – review & editing:** Rafael A. Caparros-Gonzalez, Daniel Puente-Fernandez, César Hueso-Montoro.

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
