## [Decision Letter · Decision Letter 0]

23 Oct 2024

PONE-D-24-37304Highly stressful global event affecting health sciences students: a Longitudinal Qualitative StudyPLOS ONE

Dear Dr. Salazar Granizo,

Thank you for submitting your manuscript to PLOS ONE. After careful consideration, we feel that it has merit but does not fully meet PLOS ONE’s publication criteria as it currently stands. Therefore, we invite you to submit a revised version of the manuscript that addresses the points raised during the review process.

We look forward to receiving your revised manuscript.

Kind regards,

Leona Cilar Budler

Academic Editor

PLOS ONE

Journal Requirements:

“National University of Chimborazo (doctoral studies grant 0250-CU-UNACH-SE-ORD-17-08-2022).”

4. Please note that funding information should not appear in the Acknowledgments section or other areas of your manuscript. We will only publish funding information present in the Funding Statement section of the online submission form. Please remove any funding-related text from the manuscript. 

5. We note that you have indicated that there are restrictions to data sharing for this study. For studies involving human research participant data or other sensitive data, we encourage authors to share de-identified or anonymized data. However, when data cannot be publicly shared for ethical reasons, we allow authors to make their data sets available upon request. For information on unacceptable data access restrictions, please see http://journals.plos.org/plosone/s/data-availability#loc-unacceptable-data-access-restrictions. 

6. In this instance it seems there may be acceptable restrictions in place that prevent the public sharing of your minimal data. However, in line with our goal of ensuring long-term data availability to all interested researchers, PLOS’ Data Policy states that authors cannot be the sole named individuals responsible for ensuring data access (http://journals.plos.org/plosone/s/data-availability#loc-acceptable-data-sharing-methods).

Reviewers' comments:

Reviewer's Responses to Questions

**Comments to the Author**

1. Is the manuscript technically sound, and do the data support the conclusions?

Reviewer #1: Yes

Reviewer #2: Partly

2. Has the statistical analysis been performed appropriately and rigorously? 

Reviewer #1: Yes

Reviewer #2: I Don't Know

3. Have the authors made all data underlying the findings in their manuscript fully available?

Reviewer #1: No

Reviewer #2: No

4. Is the manuscript presented in an intelligible fashion and written in standard English?

Reviewer #1: Yes

Reviewer #2: Yes

5. Review Comments to the Author

Reviewer #1: Timely topic and very good structured methodology for an qualitative study. But Figure one is not clear at all for reading. Rest of the results are readable but better to arrange in a easy readable format.

Reviewer #2: Thank you for the opportunity to review this manuscript. There was a huge amount of data and interesting approach presented here.

Overall, I found the claim for it to be qualitative research challenging. Three validated instruments were presented and 4 open ended questions (173). I didn't get a sense in the data collection or data analysis where the 3 validated instruments were situated. The hermeneutic method described as the core methodology was centered on 4 open ended questions. I wonder what the purpose of the instruments were when the results were not reported? Can more detail be given to the approach used for the volume of data (responses) analysised using ATLAS.ti? How did the codes become numerical data? I am unsure of the process of moving from ATLAS.ti to SPSS- perhaps it is my lack of understanding of this approach. 4 themes are set out in the abstract however it is not so evidence how these were developed in the analysis/ results sections. Described as topic not theme is results.

The timeframe of the data collection varied (191-192) was different to 128

Line 144 states questionnaires completed online and 192 in writing.

There are typos noted in the manuscript that need to be addressed (for example 578 the)

The discussion seemed to divert away from the goal of a highly stressful event and discuss more the specifics of the pandemic. It would be valuable to discuss within a wider context.

6. PLOS authors have the option to publish the peer review history of their article (what does this mean? ). If published, this will include your full peer review and any attached files.

**Do you want your identity to be public for this peer review?** For information about this choice, including consent withdrawal, please see our Privacy Policy .

Reviewer #1: No

Reviewer #2: No

---

## [Author Response · Author response to Decision Letter 1]

11 Nov 2024

Ecuador, November 11th, 2024

Dear

Editors,

PLOS ONE

We, the authors of the manuscript "Highly stressful global event affecting health sciences students: a Longitudinal Qualitative Study," extend our sincere gratitude for your review and valuable suggestions, which have substantially enhanced our work. We have addressed and expanded upon the requested information as follows.

Journal Requirements

Comment: Please ensure that your manuscript meets PLOS ONE's style requirements, including those for file naming.

Response: The manuscript has been thoroughly reviewed to ensure full compliance with the style requirements of this prestigious journal.

Comment: Please provide additional details regarding participant consent. In the ethics statement in the Methods and online submission information, please ensure that you have specified what type you obtained (for instance, written or verbal, and if verbal, how it was documented and witnessed). If your study included minors, state whether you obtained consent from parents or guardians. If the need for consent was waived by the ethics committee, please include this information.

Response: Thanks for the suggestion, it is expanded, and new bibliography is included:

A structured digital form, implemented in the SICOA system, facilitated the acquisition of digital informed consent from each participant, adapting the traditional written process to an online format due to the social distancing measures in effect at the time. Participants indicated their consent or non-consent through mandatory selection of a checkbox, which automatically recorded the timestamp of selection and stored it in a secure database.

If students provided consent, they proceeded to complete the research forms. The study excluded participants under 18 years of age.

Lines: 158-165

Comment: Thank you for stating the following financial disclosure: “National University of Chimborazo (doctoral studies grant 0250-CU-UNACH-SE-ORD-17-08-2022).”

Please note that funding information should not appear in the Acknowledgments section or other areas of your manuscript. We will only publish funding information present in the Funding Statement section of the online submission form. Please remove any funding-related text from the manuscript.

Response: Thank you for your suggestion:

The following statement has been added to the cover letter: “The funders had no role in study design, data collection and analysis, decision to publish, or preparation of the manuscript”.

All funding information has been removed from the manuscript.

Comment: 1. In this instance it seems there may be acceptable restrictions in place that prevent the public sharing of your minimal data. However, in line with our goal of ensuring long-term data availability to all interested researchers, PLOS’ Data Policy states that authors cannot be the sole named individuals responsible for ensuring data access (http://journals.plos.org/plosone/s/data-availability#loc-acceptable-data-sharing-methods).

Response: Thank you for your clarification:

The anonymized database has been included in: https://figshare.com/s/dc53fc6cf768800383d5

Reviewer 1

Comment: Timely topic and very good structured methodology for a qualitative study. But Figure one is not clear at all for reading. Rest of the results are readable but better to arrange in a easy readable format.

Response: Thank you for your valuable input and suggestion.

Figure 1 has been modified and restructured to enhance readability and visual clarity

Complemental material.

Line: 350

Reviewer 2

Comment: Thank you for the opportunity to review this manuscript. There was a huge amount of data and interesting approach presented here.

Overall, I found the claim for it to be qualitative research challenging.

Response: We appreciate your valuable observation regarding the methodological nature of our study.

This research is fundamentally qualitative, employing a hermeneutic analytical-interpretative approach. The primary objective was to explore and understand the experiences and perspectives of health sciences students during and after a highly stressful event, the COVID-19 pandemic, through a structured online questionnaire incorporating eight open-ended questions that enabled rich and detailed written text responses online. The hermeneutic method facilitated an in-depth interpretation of participants' narratives and experiences, obtained from their online responses.

The presence of quantitative data, extracted from students' narratives (codes), enabled complementary analysis to explore patterns in our qualitative findings, specifically regarding the relationship between numerically coded responses and demographic variables. The statistical analysis was limited to examining associations between numerical codes (coded student narratives) and demographic variables, thus rendering the quantitative component auxiliary and entirely derived from the primary qualitative analysis

Comment: Three validated instruments were presented and 4 open ended questions (173). I didn't get a sense in the data collection or data analysis where the 3 validated instruments were situated. The hermeneutic method described as the core methodology was centered on 4 open ended questions. I wonder what the purpose of the instruments were when the results were not reported?

Response: We appreciate the reviewer's thorough observation regarding the ambiguity generated in the methods section concerning data collection instruments, and we clarify as follows:

This article specifically reports the qualitative findings derived from the hermeneutic analysis of responses to eight open-ended questions, which constitute a complementary methodological component within the broader framework of the research project "Lifestyle and academic stress in Health Sciences students in an Ecuadorian educational environment." The mention of the three validated instruments ((1) Nola Pender's Lifestyle Profile Questionnaire; (2) the Systemic Cognitive Inventory for the Study of Academic Stress; and (3) the Perceived Stress Scale) does not directly correspond to the data analyzed in this specific manuscript.

To ensure methodological precision and avoid confusion, we have modified the initial paragraph of the instruments section, focusing exclusively on the qualitative instrument employed for this particular analysis. This modification allows for better correspondence between the described methodology and the presented results, maintaining the necessary rigor and coherence in the scientific report. The paragraph has been modified as follows:

To collect data, a questionnaire was created with eight open-ended questions that were previously validated by specialists with recognized teaching and research profiles in the field of health and aimed to understand health sciences students' perspectives on their lifestyles and level of academic stress in a stressful situation.

Lines: 180-183

Comment: Can more detail be given to the approach used for the volume of data (responses) analysised using ATLAS.ti? How did the codes become numerical data? I am unsure of the process of moving from ATLAS.ti to SPSS- perhaps it is my lack of understanding of this approach.

Response: We appreciate your inquiry regarding our data analysis process:

To analyze the volume of responses (n=1,735 participants; 8 questions; 2 time points), we implemented a systematic analytical approach that maintained qualitative rigor while effectively managing the extensive dataset, as follows:

ATLAS.ti Analysis: Initial Coding Phase:

• All textual responses were imported into ATLAS.ti

• Data were organized by time points (T1 and T2) and by question

• Coding was conducted following qualitative content analysis principles

• Codes were iteratively refined through research team discussions

• Code groups were established based on emerging themes

• Network views were created to visualize relationships between codes

Code Analysis and Quantification Process:

• Codes were analyzed independently for each question and assigned a number according to their presentation by question and time

SPSS Transition:

• Code numbers were organized by question and time and appended to the demographic database, matching responses by participant according to the assigned anonymized code

• Chi-square tests were conducted to analyze relationships between:

o Code presence in responses by question and theme

o Academic programs

o Participants' sex

o

This approach maintained the integrity of the qualitative analysis while enabling pattern exploration through statistical analysis, complementary to the primary qualitative findings, by identifying possible relationships between coded responses and demographic variables, as evidenced in the tables presented in the results section.

Lines: 286-311

Comment: 4 themes are set out in the abstract however it is not so evidence how these were developed in the analysis/ results sections. Described as topic not theme is results.

Response: We appreciate your observation:

In the results section, we previously used the term 'topic' instead of 'theme.' Following your valuable suggestion, we have unified the terminology to maintain clarity. Each theme has been developed in the results section through:

• Unified description of participants' experiences

• Direct participant quotations

• Code frequency analysis

• Pattern analysis across different academic programs and by gender

• Comparison between T1 and T2 periods

Lines: 185, 256, 271, 336, 341, 351, 421, 492, 570.

Comment: The timeframe of the data collection varied (191-192) was different to 128

Response: Thank you for your observation. We provide the following clarification:

The time periods (T1) April to August, 2022 and (T2) November, 2022, and March, 2023, represent the academic terms during which participants were enrolled at the university, which was a key inclusion criterion for the study.

Between August-September 2023 and November 2023-February 2024, we collected the data after obtaining ethical approval and institutional authorization. Therefore, we have included the following revised text:

Data collection involved students from six academic programs in the Faculty of Health who were enrolled during two distinct periods: Time 1 (T1): Students enrolled April-August 2022, data collection period: August-September 2023 Time 2 (T2): Students enrolled November 2022-March 2023, data collection period: November 2023-February 2024. All data were collected through a structured digital questionnaire with open-ended questions administered via the university's Academic Control System (SICOA), obtaining online narrative responses.

Lines: 215-221

Comment: Line 144 states questionnaires completed online and 192 in writing.

Response: Thank you for your important observation.

Data collection was conducted through a structured digital questionnaire administered via the university's Academic Control System (SICOA), implemented online due to social distancing measures still in effect at the time of data collection. Students wrote their narrative responses directly in the text fields. Therefore, we have included the following text:

All data were collected through a structured digital questionnaire with open-ended questions administered via the university's Academic Control System (SICOA), obtaining online narrative responses.

Lines: 219-221

Comment: There are typos noted in the manuscript that need to be addressed (for example 578 the)

Response: Thank you for your thorough review. The entire document has been revised and all typographical corrections have been incorporated into the manuscript.

Lines: 4, 5, 324, 678

Comment: The discussion seemed to divert away from the goal of a highly stressful event and discuss more the specifics of the pandemic. It would be valuable to discuss within a wider context.

Response: We appreciate your valuable observation regarding the scope of our discussion. The pandemic context, as a global stressor event, has triggered various problems among health sciences students.

We have considered your valuable comment and expanded the discussion as follows:

For the present longitudinal qualitative study, the hermeneutic method was used to delve into the perspectives and experiences of the students in a particular context and time, the COVID-19 pandemic, widely recognized as a life-threatening global stressor [26, 27, 28, 29], has particularly impacted university students through lifestyle restructuring and social limitations, affecting their mental health and subjective well-being by disrupting age-appropriate development and environmental conditions [30].

The inferential statistical analysis allowed us to examine the relationships between the variables of academic program and sex in at both time points. The findings revealed differences between male and female students in terms of the challenges faced during and after the COVID-19 pandemic, a highly stressful event.

The social limitations imposed to reduce the spread of the virus generated stressors such as the change to virtual education and social isolation, which caused a decrease in physical activity and sedentary lifestyle, eating problems these findings coincide with those of Gallé et al., who reported an increase in sedentary behavior and a decline in physical activity during this period [31]. Female students were affected by lifestyle changes and increased stress.

Lines: 640-655

Additional bibliography supporting these points has been integrated into both the manuscript text and this document.

26. Bridgland VME, Moeck EK, Green DM, Swain TL, Nayda DM, Matson LA, et al. Why the COVID-19 pandemic is a traumatic stressor. PLoS ONE [Internet]. 2021 Jan 11;16(1): e0240146. Available from: https://doi.org/10.1371/journal.pone.0240146

27. Patil S, Thute P. Mental health amidst COVID-19: A review article. Cureus [Internet]. 2022 Dec 28; Available from: https://doi.org/10.7759/cureus.33030

28. Olff M, Primasari I, Qing Y, Coimbra BM, Hovnanyan A, Grace E, et al. Mental health responses to COVID-19 around the world. European Journal of Psychotraumatology [Internet]. 2021 Jan 1;12(1). Available from: https://doi.org/10.1080/20008198.2021.1929754

29. Leung YT, Khalvati F. Exploring COVID-19–Related Stressors: Topic Modeling study. Journal of Medical Internet Research [Internet]. 2022 Jun 21;24(7): e37142. Available from: https://doi.org/10.2196/37142

30. Egbert AR, Karpiak S, Havlik R, Cankurtaran S. Global occurrence of depressive symptoms during the COVID-19 pandemic. Journal of Psychosomatic Research [Internet]. 2023 Jan 13; 166:111145. Available from: https://doi.org/10.1016/j.jpsychores.2022.111145

Lines: 889 - 905

We reiterate our gratitude for the valuable observations that have allowed us to substantially improve the clarity and precision of the manuscript.

Sincerely,

The Authors.

---

## [Decision Letter · Decision Letter 1]

20 Aug 2025

Highly stressful global event affecting health sciences students: a Longitudinal Qualitative Study

PONE-D-24-37304R1

Dear Dr. Salazar Granizo,

We’re pleased to inform you that your manuscript has been judged scientifically suitable for publication and will be formally accepted for publication once it meets all outstanding technical requirements.

Kind regards,

Leona Cilar Budler

Academic Editor

PLOS ONE

Additional Editor Comments (optional):

Please read all comments and resolve some minor issues listed by reviewer. Also, check journal guidelines to improve paper quality.

Reviewers' comments:

Reviewer's Responses to Questions

**Comments to the Author**

1. If the authors have adequately addressed your comments raised in a previous round of review and you feel that this manuscript is now acceptable for publication, you may indicate that here to bypass the “Comments to the Author” section, enter your conflict of interest statement in the “Confidential to Editor” section, and submit your "Accept" recommendation.

Reviewer #2: All comments have been addressed

Reviewer #3: All comments have been addressed

2. Is the manuscript technically sound, and do the data support the conclusions?

Reviewer #2: Yes

Reviewer #3: Yes

3. Has the statistical analysis been performed appropriately and rigorously? 

Reviewer #2: N/A

Reviewer #3: N/A

4. Have the authors made all data underlying the findings in their manuscript fully available?

Reviewer #2: Yes

Reviewer #3: Yes

5. Is the manuscript presented in an intelligible fashion and written in standard English?

Reviewer #2: Yes

Reviewer #3: Yes

6. Review Comments to the Author

Reviewer #2: Comments have been addressed and this is now a much clearer manuscript. Well done and I believe it can now move to publication

Reviewer #3: The proposed longitudinal qualitative design is well-suited to the study’s stated aims. The protocol demonstrates technical rigor, with a clearly defined target population, a robust longitudinal data collection strategy, and an analytic framework capable of capturing temporal changes in participants’ experiences. By enabling an in-depth exploration of the impacts of the global event on health sciences students over time, the methods provide the capacity to test the study’s underlying assumptions and to generate substantive, valuable insights for the field.

7. PLOS authors have the option to publish the peer review history of their article (what does this mean? ). If published, this will include your full peer review and any attached files.

**Do you want your identity to be public for this peer review?** For information about this choice, including consent withdrawal, please see our Privacy Policy .

Reviewer #2: No

Reviewer #3: No

---

## [Editor Report · Acceptance letter]

PONE-D-24-37304R1

PLOS ONE

Dear Dr. Salazar Granizo,

I'm pleased to inform you that your manuscript has been deemed suitable for publication in PLOS ONE. Congratulations! Your manuscript is now being handed over to our production team.

Kind regards,

on behalf of

Dr. Leona Cilar Budler

Academic Editor

PLOS ONE